


# Dynamic analysis of drought propagation in the context of climate change and watershed characterization: a quantitative study based on GAMLSS and Copula models

Min Li[1,2,*], Zilong Feng[1], Mingfeng Zhang[3], Lijie Shi[1], Yuhang Yao[1]

[1] College of Hydraulic Science and Engineering, Yangzhou University, Yangzhou 225000, China

[2] Key Laboratory of Flood & Drought Disaster Defense, the Ministry of Water Resources, Nanjing 210029, China

[3] Guangxi Hydraulic Research Institute, Nanning, China

*Correspondence to*: Min Li, (limintju@126.com)

**Abstract:** The analysis of the law of drought propagation under a changing environment is of great significance for drought early warning and reducing social and economic losses. Currently, few studies have analyzed the effects of meteorological factor and watershed characteristics on drought propagation based on non-stationary drought indices. In this paper, the probabilities and thresholds of meteorological drought to hydrological drought propagation were calculated using the non-stationary drought index constructed using the Generalized Additive Model for Location, Scale, and Shape (GAMLSS) model and the Copula function to assess the influence of large-scale climatic indices, meteorological elements, and watershed characteristics on the propagation characteristics of seasonal droughts. The results showed that non-stationary drought indices that incorporate meteorological factors tended to have better performance than standardized drought indices. Under the combined influence of large-scale climatic indices, temperature, specific humidity, and wind speed, the propagation probabilities became larger especially during spring and winter in the upstream and midstream regions, with the propagation thresholds in winter significantly increasing by 0.1-0.2. These mean that hydrologic droughts are more likely to be triggered. Furthermore, watershed characteristics also be factors influencing spatial differences in drought propagation.

**Keywords:** Climate change; Watershed characteristics; Drought propagation; Luanhe River basin

## 1. Introduction

As one of the major climate problems, meteorological drought poses a serious threat to the ecological



environment and social economy (Wang et al. 2022; Hao et al. 2019; Kumar et al. 2019). In a drought event,
meteorological drought often occurs first and insufficient precipitation leads to hydrological drought or
agricultural drought through the hydrological cycle (Han et al. 2019; Zhang et al. 2022; Zhong et al. 2020). This
evolution from one drought to another is called drought propagation (Zhang et al. 2021; Wossenyeleh et al. 2021;
Apurv and Cai 2020; Jehanzaib et al. 2020). After suffering from numerous drought disasters, it is widely
recognized that the impact of drought on human life can be reduced by investigating the propagation of droughts.
(Pandey et al. 2022; Dehghani et al. 2019; Le et al. 2016).
Drought is often studied based on drought indices, and the choice of drought index is crucial for
characterizing regional drought (Mahmoudi et al. 2019; Tao et al. 2021; Xu et al. 2021). Some drought indices:
the Standardized Precipitation Index (SPI), the Standardized Precipitation Evapotranspiration Index (SPEI), the
Standardized Runoff Index (SRI) and the Standardized Soil moisture Index (SSI) are used to describe the
drought characteristics of a region (Mckee et al. 1993; Vicente-Serrano et al. 2010; Shukla and Wood 2008; Xu
et al. 2021).In recent years, scholars have made a lot of efforts to examine drought propagation characteristics,
employing a wide range of analytical tools including both statistical analyses and model simulations. such as the
Copula models (Wu et al. 2022; Wang et al. 2022; Guo et al. 2020), Markov (Yeh and Hsu 2019; Vorobevskii et
al. 2022), and Variable Infiltration Capacity (VIC) model (Bhardwaj et al. 2020; Lilhare et al. 2020). Wang et al.
(2022) analyzed the propagation probability characteristics of meteorological drought to hydrological drought in
the Yiluo River Basin based on the copula function. Sattar et al. (2020) assessed the propagation probability of
meteorological drought to different categories of hydrological drought in the Han River basin using Markov
Bayesian Classifier and conditional probabilities. Bhardwaj et al. (2020) assessed drought propagation
characteristics in India based on the SPI and VIC models.
Some studies have shown that under the dual influence of climate change and human activities, the
spatiotemporal evolution characteristics of drought are difficult to analyze (Wu et al. 2022; Jehanzaib et al. 2020;
Zhou et al. 2019). Therefore, scholars analyzed the factors that affect the propagation of droughts around the
world (Li et al. 2019b). For instance, Jehanzaib et al. (2020)and Peña-Gallardo et al. (2019)have found that
climate type, climate change, catchment characteristics, and other factors can affect the propagation of drought.
Ding et al. (2021) showed the effect of climate on drought propagation by comparing the differences in
propagation time from meteorological drought to hydrological drought in different climatic regions of China.
Guo et al. (2021) assessed the impact of large reservoirs on propagation by comparing differences in drought
propagation characteristics before and after reservoir construction.



Under the influence of factors such as climate change and human activities, precipitation and runoff series show significant non-stationarity and uncertainty, and drought studies have become more complex and urgent (Wang et al. 2015; Wang et al. 2020; Jehanzaib et al. 2023). Therefore, researchers incorporate non-stationarity into drought studies through more appropriate analytical tools, The GAMLSS model is one of the commonly used methods. Previously, researchers mostly used the non-stationary drought index constructed based on the GAMLSS model to assess the impacts of climate change, human activities, and other factors on a single drought, indicating that the non-stationary drought indices have a better performance than the stationary drought index in drought research (Shao et al. 2022; Wang et al. 2023). Since then, the non-stationary drought indices have been gradually applied to the study of drought propagation. Das et al. (2022) constructed non-stationary meteorological and hydrological drought indices using large-scale climatic factors and regional meteorological elements as covariates for precipitation and runoff, respectively, and assessed the impact of external drivers on drought propagation characteristics. Overall, fewer studies incorporate non-stationary drought indices into drought propagation.

As the main source of water supply for the Beijing-Tianjin-Tangshan area, the Luanhe River Basin is responsible for multiple tasks such as urban water supply, and industrial and agricultural water supply. Frequent droughts in recent years have not only affected the supply of regional water resources but also had a serious impact on the ecological environment. Therefore, an in-depth understanding of the evolution pattern and impact mechanism of drought is of great significance to the rational allocation of water resources and sustainable development of the basin. According to some recent studies, there are nonstationary characteristics in the precipitation series and the runoff series of the Luanhe River Basin (Li et al. 2019a; Li et al. 2020). And the occurrence of the drought in Luanhe River Basin may be related to some large-scale climatic indices (Wang et al. 2018; Li et al. 2015; Wang et al. 2016). Previous studies on the Luanhe River Basin have focused on examining the effects of large-scale climatic factors on a single type of drought, with few assessments of the effects of large-scale climatic indices and regional meteorological elements on drought propagation (Li et al. 2015; Wang et al. 2015; Li et al. 2024).

Although some progress has been made in the study of drought propagation, there are few studies considering the impact of changing environments. Furthermore, spatial and temporal differences in drought propagation also be strongly related to watershed characteristics. To assess the impact of external drivers on drought propagation, the GAMLSS framework with climate factors as covariates and copula model were constructed to calculate the propagation probabilities and propagation thresholds from meteorological drought to





hydrological drought under stationary and non-stationary conditions in different seasons in this paper,
respectively. The effects of climate change on drought propagation were quantified at a seasonal scale, and the
impacts of watershed characteristics on drought propagation were explored.
**2. Study area and data**
The Luanhe River is the second largest river in Hebei Province, China, and its geographical location is
shown in Fig. 1(a). The area of the basin is about 44800 km$^2$, with an average width of 90km from east to west
and a length of 500km from north to south, including mountain 44070 km$^2$ and plain 810 km$^2$. There are obvious
differences in physical and geographical conditions, and the topography of the whole basin is high in the
northwest and low in the southeast.
The surface is flat and the river valley is wide and shallow in the Luanhe River basin. The climate
difference between the north and south of the Luanhe River basin is obvious. The annual mean temperature
ranges from 1 to 11°C, and the monthly mean temperature ranges from 17 to 25°C. Affected by the continental
monsoon climate, the basin has four distinct seasons of precipitation, with an average annual precipitation of
400~800mm, of which summer precipitation accounts for 67%-76% of the total annual precipitation; spring and
autumn account for about 9% and 15% respectively; and winter precipitation accounts for only about 2% (Li et
al. 2023). The climate type changes from cold temperate arid and semi-arid climate to warm temperate semi-
humid climate.
With global climate change, drought disasters in the Luanhe River Basin are becoming increasingly
frequent, causing serious losses to the region's ecology and socio-economy. According to historical records
(Chen et al. 2019; Chen et al. 2022; Li and Zhou 2016), the main drought events in the Luanhe River Basin
occurred in 1961, 1963, 1968, 1972, 1980-1984, 2000, 2007, and 2009. The cumulative economic losses caused
by drought disasters in the basin during the period from 1960 to 2010 exceeded 13 billion yuan. Under the
influence of climate change and human activities, the evolution law and propagation characteristics of drought in
the basin become more complex.
In this paper, the large-scale climatic indices (abbreviated as CI) Nino3.4, Atlantic Multidecadal Oscillation
(AMO), Southern Oscillation Index (SOI), Pacific Decadal Oscillation (PDO), Arctic Oscillation (AO), North
Atlantic Oscillation (NAO) and North Pacific (NP) data are derived from the National Oceanic and Atmospheric
Administration (NOAA) (http://www.esrl.noaa.gov/psd/data/climateindices) (1960-2014). The average monthly



precipitation, temperature, wind speed, specific humidity, evapotranspiration, and runoff datasets are available at
a    grid    resolution    of    0.25°    Lat    ×    0.25°    Lon    are    obtained    from
https://disc.gsfc.nasa.gov/datasets/GLDAS_NOAH10_M_2.0/. The grid-wise analysis is carried out at a
resolution of 0.25◦ Lat × 0.25◦ Lon over The Luanhe River that includes 58 grid points (Fig. 1(b)). Leaf area
index of 0.25◦ spatial resolution was derived from the Advanced Very High Resolution Radiometer (AVHRR)
Global Inventory Modeling and Mapping Studies (GIMMS) LAI3g version 2 (https://daac.ornl.gov/) (1981–

122  2015).


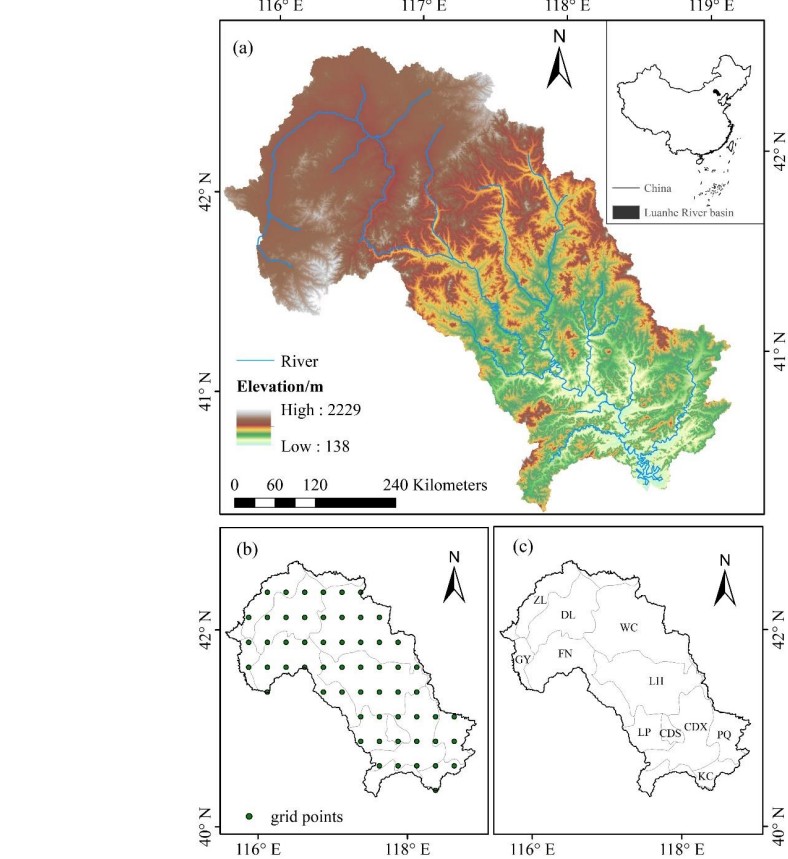

**Figure 1 The geographical location of the Luanhe River Basin (a), and the grid points contained in the**
**watershed boundaries (b), 11 subregions contained in the watershed (c)**
**3. Methods**
The current study aims to assess the impact of external drivers on drought propagation based on the




GAMLSS model, in particular, the probability and threshold of drought propagation in different seasons. Figure
3 summarizes the steps of the current study.

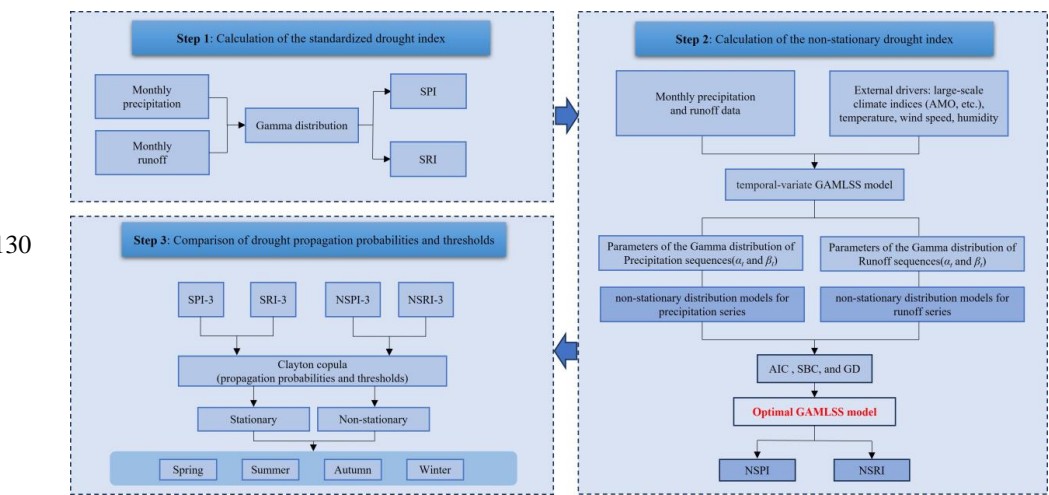

**Figure 2 Flowchart of this study**
**3.1 Pearson correlation test**
Pearson correlation test can be used to test whether there is a correlation between two sample sequences that
follow a normal distribution. The formula is as follows:
$$t = \frac{\gamma_{xy}\sqrt{n-2}}{\sqrt{1-\gamma_{xy}^2}}$$
(1)

Where n is the length of the test sample sequence, represent two different sequences, and the correlation
coefficient is calculated as follows:
$$\gamma_{xy} = \frac{\sum_{i=1}^{n}(x_i - \overline{x}) \cdot (y_i - \overline{y})}{\sqrt{\sum_{i=1}^{n}(x_i - \overline{x})^2 \cdot \sum_{i=1}^{n}(y_i - \overline{y})^2}}$$
(2)

Here, the range of $\gamma_{xy}$ values is $[-1,1]$, and when the value $|\gamma_{xy}|$ is close to 1 (Kang et al. 2022), the
correlation between the two variables is higher.
Usually, there is more than one climate factor affecting meteorological drought (Gao et al. 2020). In this
paper, Pearson correlation is used to test the correlation between large-scale climate indices and precipitation



series, to select the relevant climate variables. The wind speed, temperature, and specific humidity are
considered the main influencing factors of watershed runoff.
**3.2 The calculation of drought index**
Generalized Additive Models for Location, Scale, and Shape (GAMLSS) proposed by Rigby and
Stasinopoulos (2005)can flexibly analyze non-stationary time series, more details of GAMLSS are available in
Rigby et al. (2005). The semi-parametric additive model formula used in this study is as follows:
$$g_1(\alpha_t) = \sum_{j=1}^{j_k} h_{jk}(c_{jk}) \tag{3}$$

$$g_2(\beta_t) = \sum_{j=1}^{j_k} h_{jk}(c_{jk}) \tag{4}$$

Where $g_1(\alpha_t)$ is the link function, which is determined by the domain of the statistical parameter, namely,
if the domain of the distributed parameter $\alpha_t$ is $\alpha_t \in R$, the link function is $g_1(\alpha_t) = \alpha_t$, if $\alpha_t > 0$, then
$g_1(\alpha_t) = \ln \alpha_t$. The $h_{jk}$ represents the dependence function of the distribution parameters on the covariates
$c_{jk}$. The parameter coefficients and model residuals are estimated by RS algorithm, and whether the model
residuals approximately satisfy the normal distribution is analyzed, and the optimal fitting distribution is selected
by the AIC (Akaike information criterion), SBC (Schwarz Bayesian Criterion), and GD (Global Deviance).
**3.2.1 Stationary Model**
Taking precipitation as the object and based on the principle of hydrological calculation and normal
standardization method, SPI has the advantages of convenient data collection, relatively simple calculation,
suitable for multi-spatiotemporal scale calculation. Suppose that the precipitation series $x$ at a certain time scale
satisfies the probability density function of Gamma distribution $f(x)$:
$$f(x) = \frac{x^{\alpha-1} e^{-x/\beta}}{\beta^\alpha \Gamma(\alpha)} \tag{5}$$

In the formula, $\alpha$ and $\beta$ are scale and shape parameters ($\alpha > 0$, $\beta > 0$) and they are treated as constants in
the GAMLSS framework. The cumulative probability of precipitation is as follows:
$$F(x) = \int_0^x f(x)dx \tag{6}$$

The corresponding SPI is obtained by normalizing the cumulative probability $F(x)$ of each item.
If $0 < F(X) \leq 0.5$:
$$k = \sqrt{\ln\left[\frac{1}{F^2(x)}\right]} \tag{7}$$





$$\text{SPI} = -k - \left( \frac{c_0 + c_1 k + c_2 k^2}{1 + d_1 k + d_2 k^2 + d_3 k^3} \right) \tag{8}$$

If $0.5 < F(X) \leq 1$:
$$k = \sqrt{\ln \frac{1}{\left[ 1 - F(x) \right]^2}} \tag{9}$$

$$\text{SPI} = k - \left( \frac{c_0 + c_1 k + c_2 k^2}{1 + d_1 k + d_2 k^2 + d_3 k^3} \right) \tag{10}$$

Here: $c_0 = 2.515517$, $c_1 = 0.802853$, $c_2 = 0.010328$, $d_1 = 1.4132788$, $d_2 = 0.189269$ and
$d_3 = 0.001308$.
As a drought index that can effectively and accurately describe the hydrological drought characteristics of
the basin, SRI can be calculated by replacing the precipitation sequence with the runoff sequence and the
calculation method of SRI is similar to that of SPI. Table 1 shows the drought class classification (Kolachian and
Saghafian 2021).
**Table 1 Drought class classification and corresponding SPI values and SRI value**

| SPI\SRI value | Class |
|---|---|
| > -0.5 | Normal |
| -0.5 to -1.00 | Mild |
| -1.00 to -1.50 | Moderate |
| -1.50 to -2.00 | Severe |
| ≤ -2.00 | Extreme |

**3.2.2 Nonstationary Model**
The non-stationary modeling is based on the study by Das et al. (2022). To better study the seasonal
characteristics of drought and capture the changes in meteorological elements caused by seasonal climate change,
this paper chooses the drought index on a 3-month time scale to analyze the propagation characteristics of
drought, and the GAMLSS model is used to construct a non-stationary model for the analysis of precipitation
and runoff changes. By incorporating large-scale climate factors as covariates, a non-stationary meteorological
drought index is constructed and used to capture the non-stationary characteristics of precipitation series in the
basin. In this paper, assuming that the precipitation series at a certain time scale satisfies the gamma function
distribution, the cumulative probability is as follows:
$$F_t(x) = \int_0^x \frac{x^{\alpha_t - 1} e^{-x/\beta_t}}{\beta^{\alpha_t} \Gamma(\alpha_t)} dx \tag{11}$$




$\alpha_t$ and $\beta_t$ are the scale and position parameters of the gamma distribution. The correlated climate
variables are selected from these large-scale climate factors (e.g., AMO, SOI, PDO, AO, NAO, and NP). The
distribution of the probability density function can be fitted by the GAMLSS framework.
To capture the non-stationary characteristics of the basin runoff sequence, the non-stationary hydrological
drought index (NSRI) was constructed. The meteorological variables (wind speed, temperature, and specific
humidity) were considered as covariates for non-stationary modeling.
**3.3 The Copula model**
In multivariate drought probability analysis, the Copula function is an effective tool for constructing
multivariate joint drought distributions with multiple characteristics based on the univariate distribution and the
linkage structure between random variables. The equation is expressed as follows:
$$C(u,v) = \varphi^{-1}(\varphi(u), \varphi(v)) \tag{12}$$

where $C(u,v)$ is the Copula function that combines two random variables, $\varphi$ is the convex function, $u$
and $v$ represent the two variables respectively. Before establishing the joint distribution, the marginal
distribution of the random variables needs to be determined, and in this study, the normal distribution is used as
the marginal distribution of the meteorological drought index and hydrological drought index series. Droughts
are usually extreme climatic events, precipitation shortages and other extreme conditions, which are statistically
manifested in the behavior of data tails. And Clayton Copula can effectively capture the tail correlation between
variables, which is especially significant in the research of drought. Therefore, Clayton Copula is used to
construct the joint distribution between meteorological drought and hydrological drought indices in this paper
(Guo et al. 2021; Zhang et al. 2022; Zhang et al. 2023). Based on the Copula model, the conditional probabilities
are calculated as follows (Liu et al. 2022):
$$P\left[W \leq v / Z \leq u\right] = \frac{P(Z \leq u, W \leq v)}{P(Z \leq u)} = 1 - \frac{w(v) - C(z(u), w(v))}{1 - z(u)} \tag{13}$$

Here $Z(z_1, z_2, \cdots, z_n)$ is the conditional variable, $W(w_1, w_2, \cdots, w_n)$ is the target variable, and $z(u)$ is
used to denote the cumulative probability of $Z \geq u$, $w(v)$ denotes the cumulative probability of $W \geq v$, and
$C(z(u), w(v))$ is the joint cumulative probability. In this paper, with the meteorological drought index as the
condition and the hydrological drought index as the target, then $P\left[W \leq v / Z \leq u\right]$ denotes the conditional
probability of occurrence of hydrological drought under different meteorological drought conditions.
The drought propagation threshold (PT) is commonly defined as the severity of the meteorological drought
that is most likely to cause hydrological drought, i.e., the SPI critical threshold. In this paper, the conditional
probability density of SPI was calculated for each scenario in the interval of -3 to 3 at an interval of 0.01, and
when SRI ≤ -0.5, the SPI value corresponding to the maximum point of the conditional probability density is the
meteorological drought threshold that triggers hydrological drought (Zhou et al. 2022).
To visualize more intuitively the difference between meteorological drought to hydrological drought
propagation thresholds under non-stationary and stationary condition, the change rate of drought propagation
thresholds was calculated with the following equations:


$$R_c = \left| \frac{T_n - T_s}{T_s} \right| \times 100\% \tag{14}$$

where $T_n$ and $T_s$ the thresholds of meteorological drought to hydrological drought propagation under non-
stationary conditions and stationary conditions.
**4. Results**
**4.1 Selection of Climate Indices**
To select the relevant climate variables linked with meteorological drought in the Luanhe River basin, the
Pearson correlation test was carried out to test the correlation between cumulative precipitation series at different
time scales K (K = 1, 3, 6, 12, 24 months) and the *CI* with a lead time M (M = 0, 1, 2, 3 months) for all regions
of the basin. The standardized climatic indices series have been averaged on a period (AP) of 1, 3, 6, 12, and 24
months (*CI-n*: *CI* with the AP=n month). To analyze the seasonal drought characteristics of the basin, we
selected the significant climate indices for the cumulative precipitation series on a three-month time scale.
According to the results of the correlation test, AMO-1 and AMO-24 with a lead time of M=0 months were
selected as covariates for the precipitation series and the test results are shown in Fig. 3.
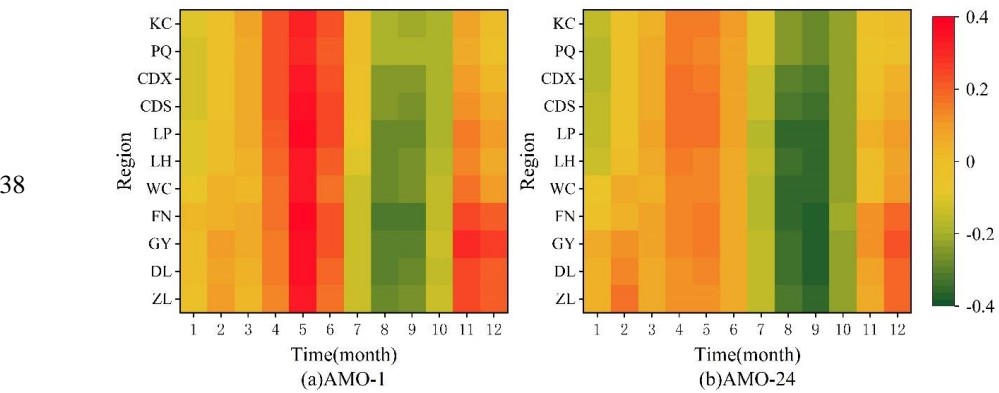
**Figure 3 The correlation between AMO and precipitation series**
Trends of temperature, wind speed, and humidity in different seasons were calculated by the Mann-Kendall
(M-K) trend analysis method in the watershed (Mann 1945; Cheng et al. 2023). The results are presented in
Table 2. When the absolute value of Z is greater than 1.96, it indicates that the series shows a significant level of
$p < 0.05$. The temperature shows a significant upward trend in four seasons. Wind speed shows a decreasing


trend in spring and summer and an increasing trend in autumn and winter. Relative humidity showed an
increasing trend in spring, summer, and winter, and a decreasing trend in summer.
**Table 2 Trends of temperature, wind speed, and specific humidity in different seasons (The bold numbers**
**represent the series shows a significant trend.)**

| | Z | | | |
|---|---|---|---|---|
| | Spring | Summer | Autumn | Winter |
| Temperature | **4.55** | **4.37** | **4.13** | **3.66** |
| Wind speed | -0.03 | **-4.21** | 0.12 | 0.58 |
| Specific humidity | 1.29 | -0.07 | 1.10 | **2.61** |

**4.2 Preference of GAMLSS model**
**4.2.1 The simulation of precipitation series**
GAMLSS framework was used to model the precipitation in each region of the watershed. To analyze the
seasonal drought characteristics of the region, the SPI was calculated for 3-month time scales in this article.
According to the correlation test results, AMO (AP=1 and AP=24) was selected as the significant *CI* for non-
stationary modeling of precipitation. Seven different situations were considered according to the structure of the
GAMLSS model (the model types are shown in Table 3). The AIC, SBC, and GD were used to select the optimal
model, taking the CDS region as an example, the results of model preferences for the precipitation series are
shown in Table 3.
**Table 3 Different model situations considered for precipitation simulation (CI-n: CI with the AP=n month)**

| Model | Parameters | |
|---|---|---|
| | $\alpha_t$ | $\beta_t$ |
| Mod 1 | ~1 | ~1 |
| Mod 2 | ~1 | ~ AMO-1, AMO-24 |
| Mod 3 | ~ AMO-1, AMO-24 | ~1 |
| Mod 4 | ~1 | ~ AMO-24 |
| Mod 5 | ~ AMO-24 | ~1 |
| Mod 6 | ~1 | ~ AMO-1 |
| Mod 7 | ~ AMO-1 | ~1 |

**Table 4 AIC, SBC, and GD of the different models of precipitation in the CDS region (the Bold indicates**
**the optimal model)**

| Model | Spring | | | Summer | | | Autumn | | | Winter | | |
|---|---|---|---|---|---|---|---|---|---|---|---|---|
| | AIC | SBC | GD | AIC | SBC | GD | AIC | SBC | GD | AIC | SBC | GD |
| Mod 1 | 498.2 | 502.2 | 494.2 | 626.8 | 630.7 | 622.8 | **505.8** | **509.7** | **501.8** | 325.3 | 329.3 | 321.3 |



| | | | | | | | | | | | | |
|---|---|---|---|---|---|---|---|---|---|---|---|---|
| Mod2 | 501.7 | 509.6 | 493.7 | 629.2 | 637.1 | 621.2 | 509.4 | 517.3 | 501.4 | **322.5** | **330.4** | **314.5** |
| Mod3 | 495.1 | 503.0 | 487.1 | 627.7 | 635.6 | 619.7 | 508.3 | 516.2 | 500.3 | 329.3 | 337.1 | 321.3 |
| Mod4 | 500.0 | 506.0 | 494.0 | 627.2 | 633.1 | 621.2 | 507.7 | 513.6 | 501.7 | 329.9 | 324.0 | 318.0 |
| Mod5 | 499.1 | 505.0 | 493.1 | **625.8** | **631.7** | **619.8** | 507.8 | 507.8 | 513.7 | 327.3 | 333.2 | 321.3 |
| Mod6 | 500.2 | 506.1 | 494.2 | 627.6 | 633.5 | 621.6 | 507.8 | 513.7 | 501.8 | 327.2 | 333.1 | 321.2 |
| Mod7 | **494.1** | **500.0** | **488.1** | 627.6 | 633.5 | 621.6 | 507.2 | 513.1 | 501.2 | 327.3 | 333.2 | 321.3 |

As can be seen from Table 4, for the non-stationary models of precipitation in the CDS region, among all
the models with climate index as covariates, Mod7 has the best performance in spring, with the AIC, SBC, and
GD of 494.1, 500.0 and 488.1 respectively. The optimal model in summer was Mod5, the AIC, SBC, and GD
were 625.8, 631.7, and 619.8. In autumn, the optimal model was Mod1, the AIC, SBC, and GD were 505.8,
509.7, and 501.8. Mod2 had the best performance in winter, with the AIC, SBC, and GD of 322.5, 330.4, and
314.5. The results of the estimated model parameters of the precipitation in the CDS region are shown in Table 5:
**Table 5 Model parameters estimation results in four seasons in the CDS region**

| Season | Parameters |
|---|---|
| Spring | $\alpha_t = \exp(4.17 + 0.13 AMO_{t\text{-}1})$<br>$\beta_t = \exp(-0.98)$ |
| Summer | $\alpha_t = \exp(5.75 - 0.10 AMO_{t\text{-}24})$<br>$\beta_t = \exp(-1.43)$ |
| Autumn | $\alpha_t = \exp(4.29)$<br>$\beta_t = \exp(0.40)$ |
| Winter | $\alpha_t = \exp(2.04)$<br>$\beta_t = \exp(-0.20 - 0.49 AMO_{t\text{-}1} + 0.29 AMO_{t\text{-}24})$ |

To assess the quality of the fitting, Fig.4 provides the simulation of precipitation from the GAMLSS
framework (Taking the CDS region as an example). As shown in Fig.4, these red dots represent precipitation
observations, light grey areas represent areas between the 5% and 95% centile curves, dark grey areas represent
areas between the 25% and 75% centile curves, and black lines represent median (50%), the black dashed line in
the worm plot of the fitted residuals indicates the 95% confidence interval.
It can be seen from Fig.44 that the precipitation data values of the four seasons were basically within the 95%
quantile interval, the deviation values in the worm chart were evenly distributed in the 95% confidence interval,




274 and there was no obvious excess, which indicated that the residual fitting of Gamma distribution meets the

275 conditions. In general, the temporal behavior associated with the data was significant, the results of the model

276 (Fig.4) seem to reproduce the behavior of the data, especially to capture the large dispersion characteristics of the

277 data.

278

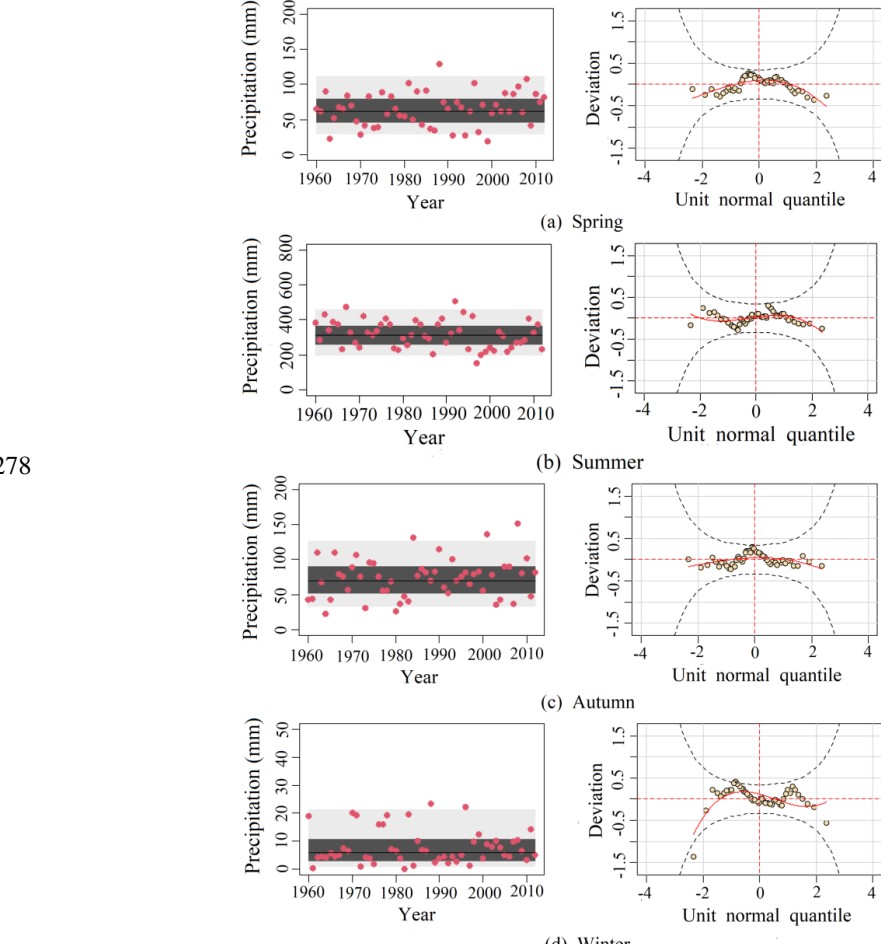

279 **Figure 4 Fitting results of four seasons of precipitation series in the CDS region**

280 **4.2.2 The simulation of the runoff series**

281  For the simulation of runoff, temperature(T), specific humidity(H), and wind speed(W) were considered as

282 covariates of the shape and position parameters of the gamma distribution. Some of the model situations

283 considered are shown in Table 6, and taking the CDS region as an example, the optimal results are listed in Table

284 7.



**Table 6 Different model situations considered for runoff simulation**

| Model | Parameter | |
|---|---|---|
| | $\alpha_t$ | $\beta_t$ |
| Mod 1 | ~1 | ~1 |
| Mod 2 | ~1 | ~ T and H |
| Mod 3 | ~ T and H | ~1 |
| Mod 4 | ~1 | ~ T |
| Mod 5 | ~ T | ~1 |
| Mod 6 | ~1 | ~ H |
| Mod 7 | ~ H | ~1 |
| Mod 8 | ~1 | ~ T, H and W |
| Mod 9 | ~ T, H and W | ~1 |

**Table 7 AIC, SBC, and GD of the best suitable model of the non-stationary model of runoff in the CDS region**

| Season | The optimal model | |
|---|---|---|
| Spring | Mod 9 | AIC: -70.28 |
| | | SBC: -60.43 |
| | | GD: -80.27 |
| Summer | Mod 4 | AIC:136.19 |
| | | SBC:142.10 |
| | | GD:130.19 |
| Autumn | Mod 3 | AIC: -58.67 |
| | | SBC: -50.79 |
| | | GD: -66.67 |
| Winter | Mod 3 | AIC: -4477 |
| | | SBC: -439.89 |
| | | GD: -455.77 |

The results of the estimated model parameters of the runoff in the CDS region as an example were shown in Table 8. As seen in Table 8, the main factors affecting the spring runoff series were temperature, specific humidity, and wind speed, with specific humidity having a greater influence than the other two factors. In summer, temperature was the main factor influencing the runoff series. In autumn and winter, runoff sequences were mainly influenced by temperature and specific humidity.

**Table 8 Model parameters estimation results in four seasons of the CDS region**





| Season | Parameter |
|---|---|
| Spring | $\alpha_t = \exp(-1.57 - 0.37T_t + 0.54H_t + 0.28W_t)$ |
| | $\beta_t = \exp(-0.42)$ |
| Summer | $\alpha_t = \exp(0.62)$ |
| | $\beta_t = \exp(-0.73 + 0.23T_t)$ |
| Autumn | $\alpha_t = \exp(-1.45 - 0.29T_t + 0.48H_t)$ |
| | $\beta_t = \exp(-0.41)$ |
| Winter | $\alpha_t = \exp(-4.85 - 1.91T_t + 1.34H_t)$ |
| | $\beta_t = \exp(0.48)$ |

The simulation results of the stationary model and non-stationary for runoff in the CDS region are shown in Fig.5. As can be seen from Fig.5, most of the runoff data values (red points) of the four seasons were located in the light gray area (5% and 95% centile curves), and the data deviations in the worm plots were evenly distributed in the 95% confidence interval (between the two black ellipse dotted lines), which show that non-stationary gamma distribution meet the requirements for the fitting of runoff series. In Fig.5, the non-stationary model showed the time variation characteristics of the runoff series flexibly. Generally, the non-stationary model can describe the variability of runoff series accurately. In summary, the non-stationary model with temperature, humidity, and wind speed were considered as covariates that can capture the time variation characteristics of the runoff series.

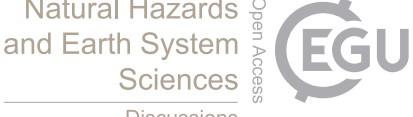



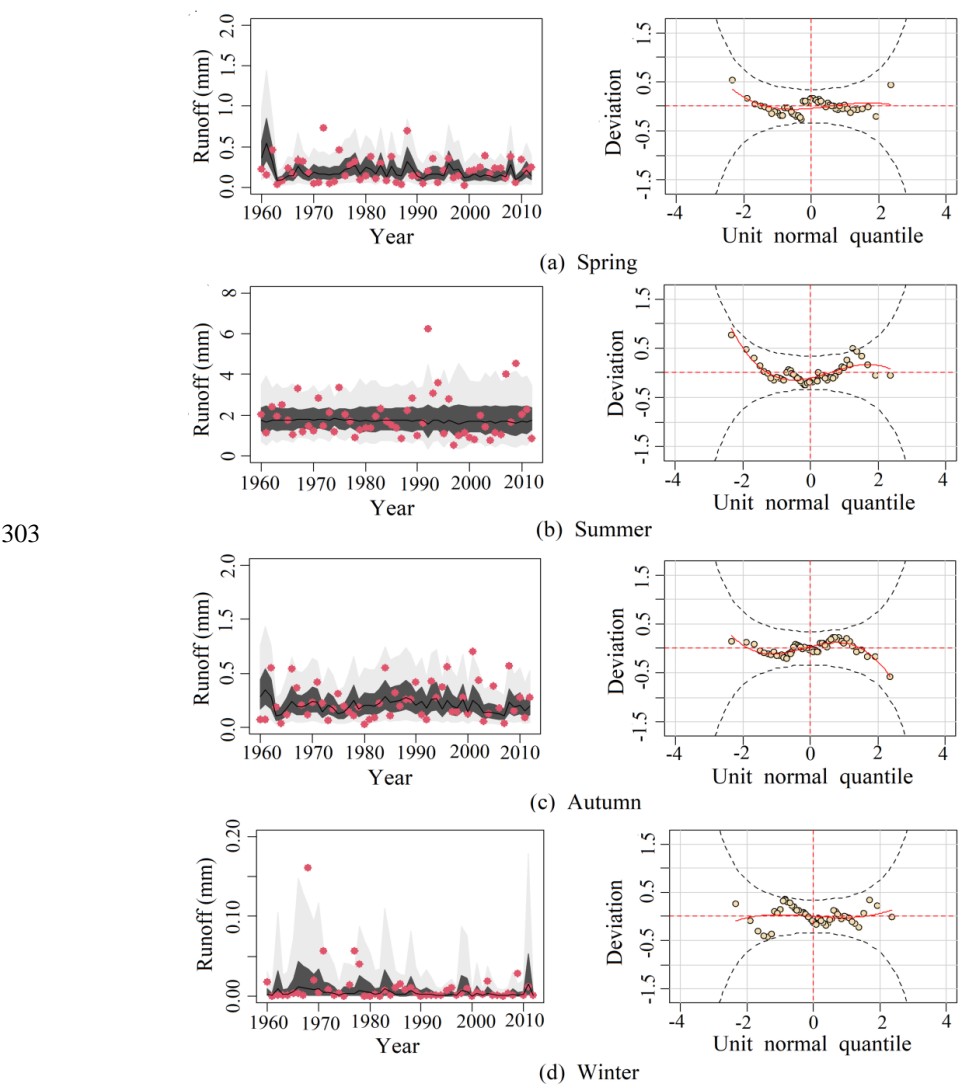

303

**Figure 5 Fitting results of four seasons of runoff series in the CDS region**

**4.3 Calculation of stationary and non-stationary indices**

According to the simulation results of the model in Section 4.2, the non-stationary models have better performance than the stationary models in the simulation of runoff series in all regions. The comparison results of SRI and NSRI in different seasons in CDS are shown in Fig.6. It can be seen that the distribution of two indices is generally similar. Furthermore, the climate factors had different impacts on the index in different seasons, with the smallest impact on summer and the most significant impact on winter.




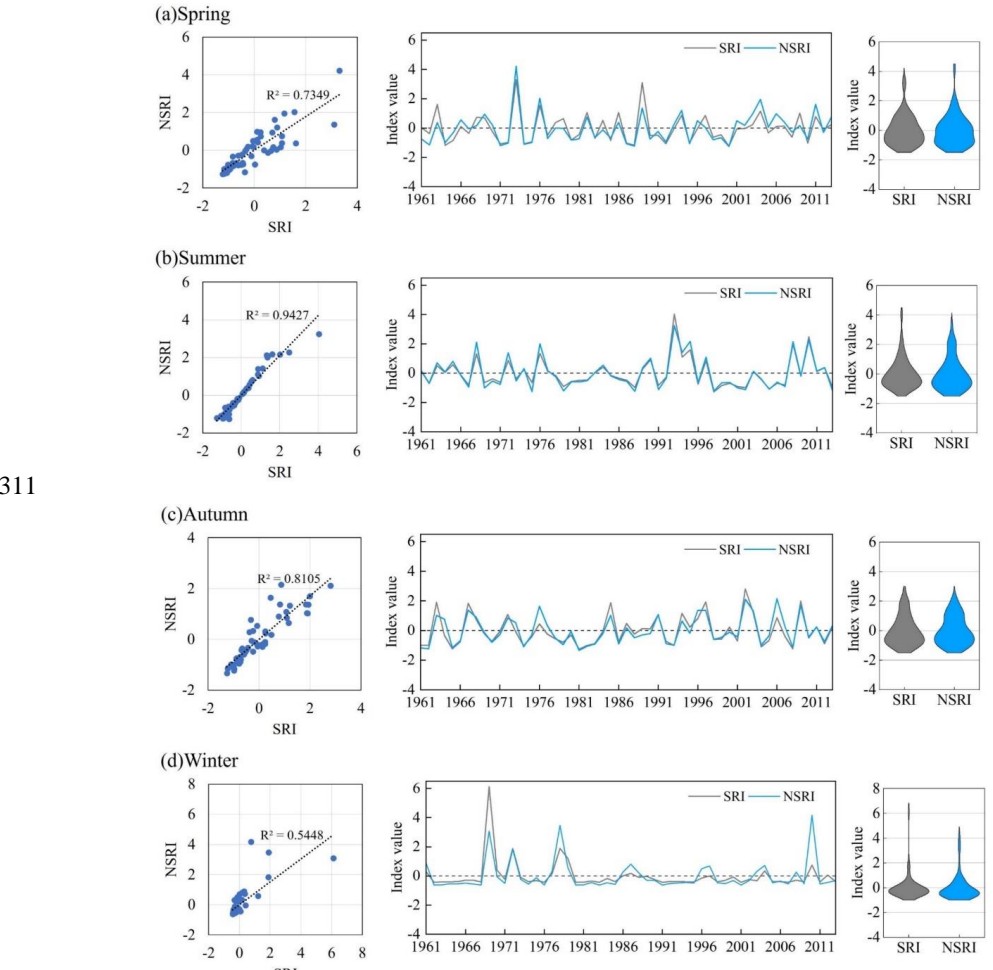

**Figure 6 Comparison of SRI and NSRI in different seasons in the CDS region (a: Spring; b: Summer; c:**
**Autumn; d: Winter)**
**4.4 Drought propagation probability**
Based on the Copula model, the probabilities of meteorological drought propagation to hydrological
drought can be calculated, and the impact of climate change on drought propagation can be analyzed. The
calculated results in different seasons and different regions were shown in Figs.7-10, where the solid and dashed
lines indicate the calculated results of the non-stationarity model and the stationarity model, respectively, and
black, red, blue, and green represent extreme drought, severe drought, moderate drought, and mild drought,
respectively. According to the analysis results in Figs.7-10, the probabilities of the occurrence of hydrological
drought increased with the decrease of SPI, and as the degree of meteorological drought worsened, it might lead
to more severe hydrological drought. In addition, the drought propagation probabilities calculated based on the
non-stationarity model were significantly different from those calculated by the stationarity model, and they also
differ in different seasons and regions.

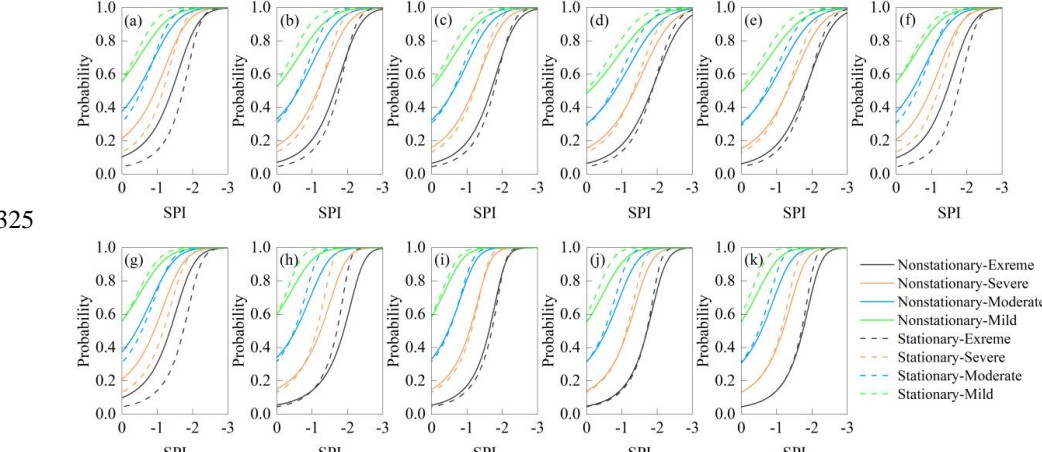


**Figure 7 Probability of drought propagation in spring for each region (a: ZL; b: DL; c: GY; d: FN; e: WC; f: LH; g: LP; h: CDS; i: CDX; j: PQ; k: KC)**

Fig.7 shows the calculated results of drought propagation probabilities in spring in 11 regions. In the
upstream (ZL, DL, GY,) and middle regions (WC, FN, LH, LP, and CDS) of the basin, the drought propagation
probabilities calculated by the non-stationary model were significantly different from those calculated by the
stationary model, while the calculated results were relatively close in the downstream areas such as CDX, PQ
and KC. For the upstream and middle regions, under the same meteorological drought conditions, the
probabilities of severe and extreme hydrological drought calculated based on the non-stationary model were
larger than that of the stationary model, while in the downstream area, the probabilities of hydrological drought
calculated by the stationary model were slightly higher than that of the non-stationary model. According to the
modeling structure of the precipitation and runoff sequence in spring in section 4.2, under the combined
influence of climatic factors AMO, temperature, specific humidity, and wind speed, regional hydrological
drought is more likely to occur. In contrast to the stationary conditions, the increase in temperature may be the
main factor that causes the hydrological drought to become more severe in spring.



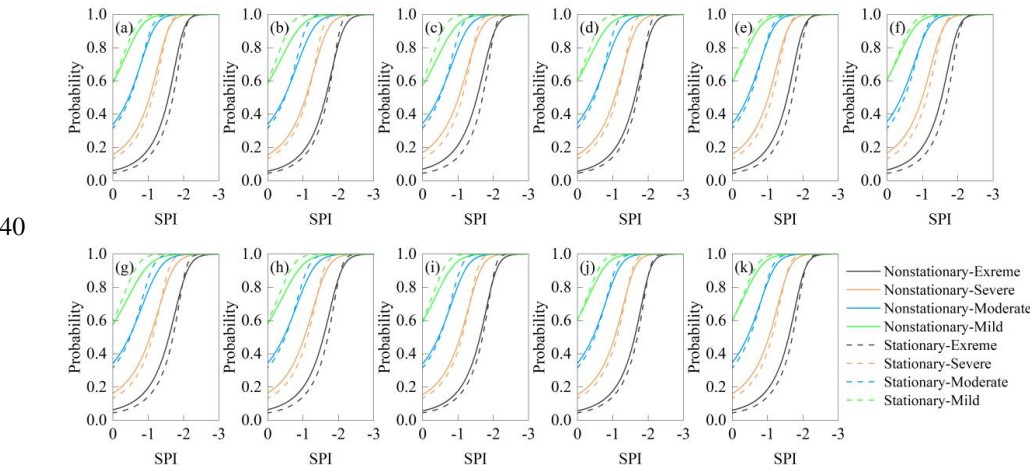


**Figure 8 Probability of drought propagation in summer for each region (a: ZL; b: DL; c: GY; d: FN; e: WC; f: LH; g: LP; h: CDS; i: CDX; j: PQ; k: KC)**

In summer (Fig.8), in each region, the difference between the drought propagation probabilities calculated by the non-stationary model and the results calculated by the stationary model was not significant, and the probability of occurrence of severe and extreme hydrological droughts calculated by the non-stationary model was larger. Taking the ZL region as an example (Fig.8(a)), when climate change was not considered, the probability of severe hydrological drought and extreme hydrological drought was 0.6 and 0.17, respectively. Under the influence of the changing environment, the probability of causing severe hydrological drought and extreme hydrological drought was 0.62 and 0.2 respectively. This means that climate changes had little impact on drought propagation in the basin during the summer when precipitation was abundant. In contrast to the stationary conditions, the AMO and temperature may be the main climate reasons for the greater probability of drought propagation in summer (Zhang et al. 2022).

Different from spring and summer, in autumn (Fig.9), The probabilities of occurrence of moderate drought and more severe hydrological droughts calculated by the non-stationary model were larger than those of the stationary model in the upstream (ZL, DL, and GY) and downstream regions (CDX, PQ, and KC), which indicated that the propagation of droughts in the upstream and downstream regions was influenced by climate change significantly. Temperature and humidity may be the main climate-influencing factors for the significant increase of the drought propagation probability in the upstream and downstream areas. Unlike the upstream and downstream areas, these climatic factors may not be the main cause of the propagation of drought in the midstream region (WC, FN, LH, LP, and CDS).

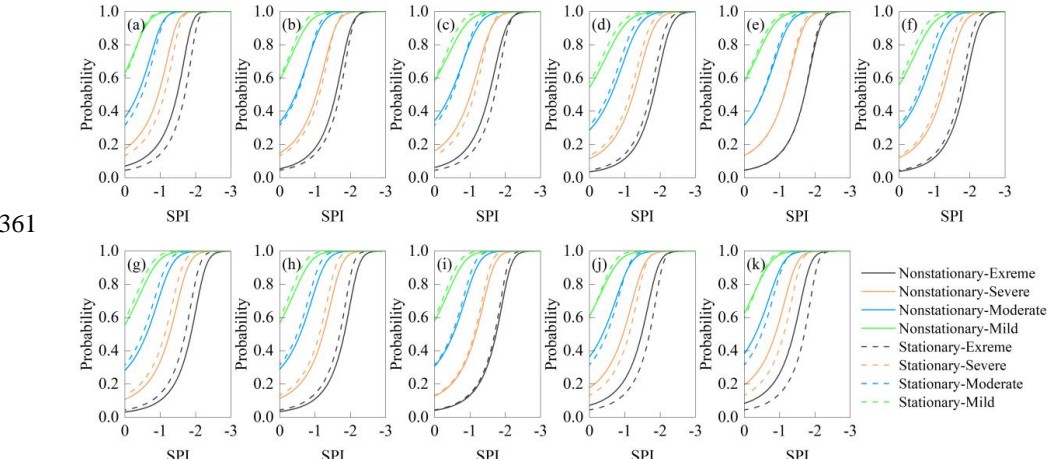

**Figure 9 Probability of drought propagation in autumn for each region (a: ZL; b: DL; c: GY; d: FN; e: WC; f: LH; g: LP; h: CDS; i: CDX; j: PQ; k: KC)**

In winter (Fig.10), the probabilities of occurrence of moderate and more severe hydrological droughts in the upstream and midstream regions calculated based on the non-stationary model were significantly larger than those calculated by the stationary model. Taking the WC station as an example, when climate change was not considered, the probabilities of occurrence of moderate, severe, and extreme hydrological droughts under moderate meteorological drought conditions were about 0.8, 0.6, and 0.4, respectively, while under the influence of environmental change, the probabilities of moderate, severe and extreme hydrological droughts were about 0.9, 0.8 and 0.6, respectively. In most of the downstream areas, the difference between the calculation results of the two models was relatively small. Under the combined influence of AMO, temperature, wind speed, and specific humidity, the probabilities of drought propagation are increased. In upstream, the decrease in wind speed may be the main climate factors affecting the occurrence of severe drought, and the increase in temperature and the decrease in specific humidity may be the main climate factors affecting the occurrence of severe drought in midstream regions. In downstream areas, these climatic factors may not be the main influences on drought propagation.


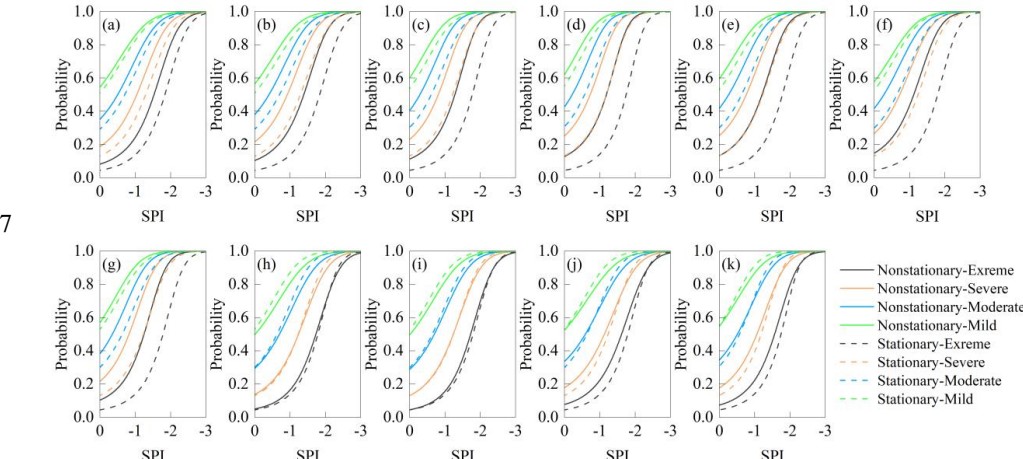

**Figure 10 Probability of drought propagation in winter for each region (a: ZL; b: DL; c: GY; d: FN; e: WC; f: LH; g: LP; h: CDS; i: CDX; j: PQ; k: KC)**
Comparing the four seasons, the probabilities of occurrence of moderate and more severe droughts were the
lowest in spring, but the highest in winter, this phenomenon was significant under non-stationarity conditions.
Taking the FN region as an example (Fig.7(d)- Fig.10(d)), the probabilities of moderate meteorological drought
propagating as moderate, severe, and extreme hydrological drought in spring under non-stationarity conditions
were close to 0.6, 0.4, and 0.15, respectively, while in winter, the probabilities of propagating as moderate,
severe and extreme hydrological drought under the same meteorological drought conditions were close to 0.9,
0.7 and 0.4, respectively. The reasons for the differences in the probabilities of drought propagation under
stationary and non-stationary conditions are complex. On the one hand, non-stationary models capture changes
caused by interannual variability, and on the other hand, they are affected by AMO, temperature, wind speed, and
relative humidity. There may be some differences in the effects of various meteorological factors on drought in
different seasons. From the results in Section 4.2, the drought propagation is affected by the combined effects of
AMO, temperature, wind speed, and relative humidity in spring, with relative humidity as the main influencing
factor. In summer, drought propagation is mainly influenced by AMO and temperature. In the fall, it is
influenced by temperature and relative humidity, with relative humidity being the main influencing factor. In
winter, it is influenced by a combination of AMO, temperature, and relative humidity, with temperature being the
most important influencing factor. Comparing the four seasons, meteorological factors have the most serious
effect on winter drought. In addition, there are some differences in the effects of meteorological factors on
drought in different regions. Temperatures show a significant upward trend, which may mean that extreme runoff


events will be more frequent. During the dry season, high temperatures increase evapotranspiration from surface
water bodies, vegetation, etc., resulting in reduced runoff and lower soil moisture content will increase the risk of
hydrological drought (Huang et al. 2017; Guo et al. 2021). Changes in humidity affect the efficiency of
evapotranspiration, and higher humidity will reduce the transfer of water from the surface and plants to the
atmosphere, limiting the development of drought. However, this effect may be limited by increased evaporation
from increasing temperatures.
**4.5 Drought propagation threshold**

405       Based on the Copula model, the thresholds that trigger hydrological droughts under stationary and non-

stationary conditions (i.e., the propagation thresholds for drought) can be calculated, the results are shown in Fig.
11. The change rate of the meteorological drought to hydrological drought propagation thresholds are shown in
Figure 12 As can be seen from Figs.11 and 12, there were obvious regional and seasonal characteristics of
drought propagation thresholds. In this paper, the higher the drought propagation thresholds, the more likely
hydrological drought is to be triggered.

411       In spring (Fig.11(a)), comparing the results of calculations based on the stationary model and the non-

stationary model, the drought propagation thresholds were the smallest in FN, WC region, and the highest values
occurred in the downstream region (CDS, CDX, PQ, KC) under the stationary condition. The distribution of
drought propagation thresholds under non-stationary conditions was similar to that under stationary conditions.
In addition, compared with the stationary condition, the drought propagation thresholds were higher in most
regions under non-stationary condition. It indicated that hydrological droughts were more difficult to be
triggered in most regions under the influence of climatic factors such as temperature, specific humidity, wind
speed, and AMO. In summer (Fig. 11(b)), There was no significant difference in drought propagation thresholds
in all regions under stationary conditions and non-stationary conditions. In autumn (Fig. 11(c)), the drought
propagation thresholds in the river basin were close to that in summer. Under stationary conditions, the drought
propagation thresholds were close to -0.55 in most regions. Comparing stationary conditions, the drought
propagation thresholds increased in ZL, PQ, and KC, while decreasing in middle-stream areas (FN, WC, LH, LP,
CDS, CDX) under non-stationary conditions. In winter (Fig. 11(d)), there were significant differences in regional
drought propagation thresholds between stationary and non-stationary conditions. Under stationary conditions,
the drought propagation thresholds of the basin were relatively lower than those in spring, summer, and autumn,
with values ranging from -0.70 to -0.65. Under non-stationary conditions, the drought propagation thresholds





increased generally, especially in the midstream region. From Fig. 12, it can be seen that drought propagation
thresholds were most affected by large-scale climate factors and meteorological factors in winter, with a rate of
change greater than 10% or even 20% in most regions, followed by spring, with the least change in the summer
and autumn seasons. It indicated that hydrological drought was more likely to occur during winter due to climate
factors.

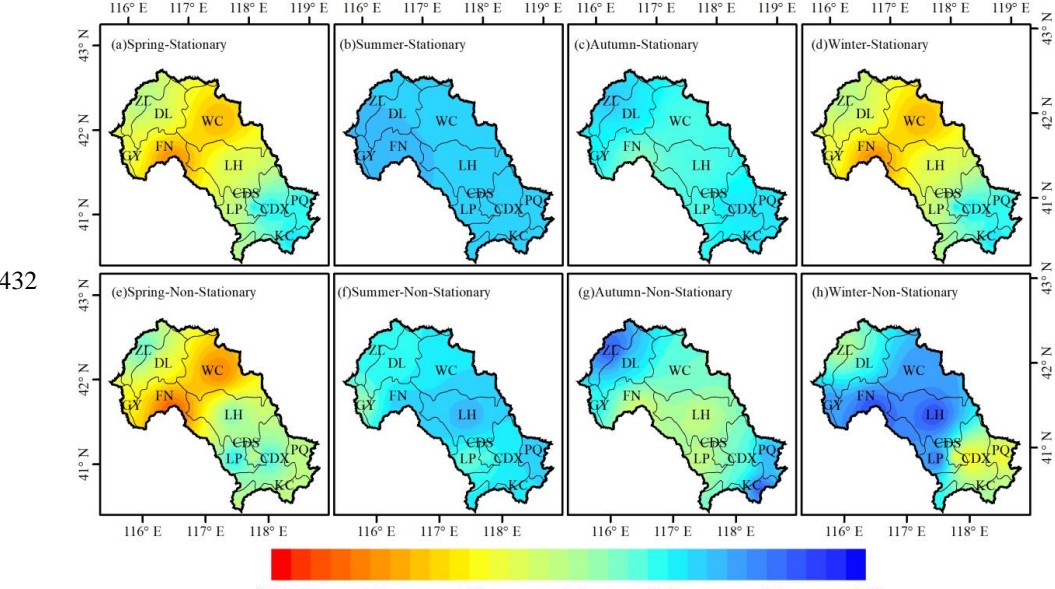


**Figure 11 Drought propagation thresholds in different seasons under stationary and non-stationary**
**conditions**


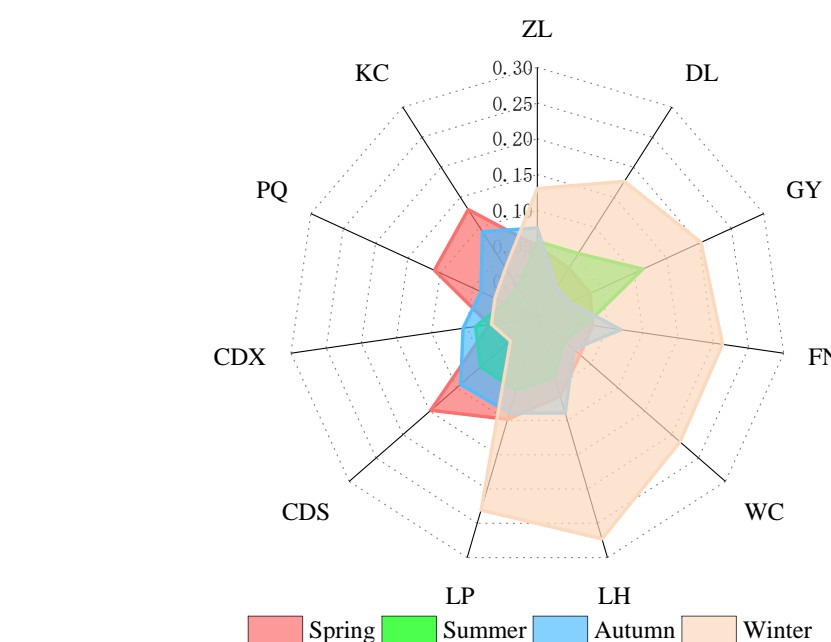


**Figure 12 The change rate of drought propagation thresholds in different seasons**


437        Comparing the four seasons, the drought propagation thresholds in most areas were relatively low in spring

and winter, and relatively high in summer and autumn under the stationary conditions. In contrast to winter and
spring, precipitation was more abundant in summer and autumn, the runoff was more sensitive to precipitation,
the propagation time from meteorological drought to hydrological drought was shorter, and a milder degree of
meteorological drought might trigger hydrological drought. However, under the influence of climatic factors, the
drought propagation thresholds of all four seasons changed. From the point of view of the model structure,
climatic factors such as AMO, specific humidity, temperature, and wind speed had an impact on the occurrence
of seasonal drought. Compared with spring, summer, and autumn, temperature and specific humidity had a great
influence on the propagation of drought in winter. The increase in temperature may be the main reason for the
occurrence of hydrological drought in winter.
**5. Discussion**

448        There are some differences in drought propagation thresholds in different regions, which may be caused by

the watershed characteristics, including slope and so on (Han et al. 2023, Liu et al. 2023, Zhou et al. 2021). To
further explore the spatial differences of propagation thresholds, the slope, average evapotranspiration, soil water
content (0-10 cm, 10-40 cm, 40-100 cm, 100-200 cm), and leaf area index in each region were calculated, and




the relationships between the propagation thresholds and the factors were explored for each region. As shown in Table 9`, these factors may be one of the reasons for the spatial differences in drought propagation thresholds. Evapotranspiration and shallow soil moisture are dominant among these factors, followed by the effects of slope and vegetation on drought propagation. The drought resistance of the watershed decreases when the slope increases and the water storage capacity decreases, and meteorological droughts are more likely to trigger hydrologic droughts. Evapotranspiration is a key part of the water cycle and directly reflects the exchange of water between soil, vegetation, and the atmosphere. There is a positive correlation between evapotranspiration and drought propagation thresholds, and an increase in evapotranspiration leads to a decrease in surface water resources, which may increase the risk of drought propagation(Guo et al. 2020; Yao et al. 2022). Soil moisture content may also be one of the factors causing spatial differences in drought propagation thresholds, with shallow soil having a greater impact on drought propagation than deep soil. Vegetation cover also affects drought propagation, and more vegetation can increase water retention in a watershed and improve its drought resistance. However, when meteorological drought is severe, vegetation in a water-starved condition will consume more water through transpiration, accelerating the onset of drought.

**Table 9 The characteristics of the study area, including slope, evapotranspiration(E), soil moisture content (0-10 cm underground) (SMC0-10cm), soil moisture content (10-40 cm underground) (SMC10-40cm), soil moisture content (40-100 cm underground) (SMC40-100cm), soil moisture content (100-200 cm underground) (SMC100-200cm), Lead area index (LAI)**

| Region | Slope | E(mm) | SMC0-10 cm | SMC10-40 cm | SMC40-100 cm | SMC100-200 cm | LAI |
|---|---|---|---|---|---|---|---|
| ZL | 2.30 | 84.76 | 42.35 | 130.30 | 183.53 | 522.55 | 0.50 |
| DL | 3.60 | 88.79 | 42.67 | 129.60 | 180.51 | 524.61 | 0.50 |
| GY | 2.36 | 90.27 | 43.43 | 132.57 | 184.80 | 531.07 | 0.56 |
| FN | 10.35 | 96.61 | 44.19 | 136.50 | 196.74 | 523.87 | 0.86 |
| WC | 10.06 | 95.28 | 42.38 | 122.27 | 171.65 | 514.83 | 1.03 |
| LH | 12.64 | 103.76 | 46.84 | 143.74 | 216.09 | 456.18 | 1.21 |
| LP | 12.48 | 110.20 | 48.25 | 149.69 | 230.37 | 477.93 | 1.08 |
| CDS | 10.27 | 112.99 | 55.99 | 157.92 | 236.12 | 612.83 | 0.81 |
| CDX | 13.04 | 113.28 | 45.93 | 143.71 | 231.90 | 394.67 | 1.31 |
| PQ | 11.59 | 114.56 | 45.30 | 135.91 | 209.45 | 516.40 | 1.02 |





| KC | 14.56 | 117.83 | 45.81 | 139.27 | 222.59 | 465.23 | 1.24 |
|---|---|---|---|---|---|---|---|
| Pearson for PT | 0.34 | 0.71 | 0.48 | 0.50 | 0.66 | -0.09 | 0.23 |

**6. Conclusions**

Many studies have pointed out that climate change and human activities significantly impact the occurrence of drought in the Luanhe River basin. In this paper, meteorological drought and hydrological drought were characterized by the SPI and SRI respectively. The drought propagation probabilities and thresholds in all seasons were calculated based on the non-stationary drought index constructed by the GAMLSS model and the Copula function, the influence of climate changeand watershed characteristics on drought propagation was analyzed. The following conclusions can be drawn.

(1) AMO-1 and AMO-24 have a significant impact on the precipitation series in the Luanhe River basin. The temperature, wind speed, and specific humidity were considered as the main influencing climate factors of the runoff series.

(2) Based on the GAMLSS framework, both the stationary model and non-stationary model have a good fitting effect on the precipitation and runoff series of the basin, but overall, the non-stationary model can capture the time variation characteristics of these series more accurately.

(3) For most regions, the probabilities of drought propagation under non-stationary conditions were greater than that under stationarity conditions. Compared to summer and autumn, spring and winter were more prone to hydrological drought and may experience more severe hydrological drought.

(4) With regard to the drought propagation thresholds, non-stationary conditions were more likely to trigger hydrological drought than stationary conditions, this phenomenon was particularly evident in the midstream and upstream regions in winter, with drought propagation thresholds increasing by 0.1-0.2 under non-stationary conditions compared to stationary conditions. The increase of temperature may be the key factors contributing to the occurrence of hydrological drought in the basin.

(5) Watershed characteristics were important factors in the spatial differences of drought propagation characteristics, including vegetation cover and so on. Among them, there was a high correlation (the absolute value of correlation coefficient > 0.5) between evapotranspiration, soil moisture content (10-40 cm underground, 40-100 cm underground) and drought propagation characteristics.



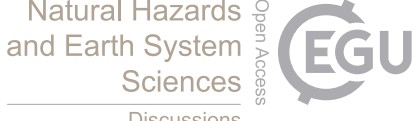

**Limitation:** There are many driving factors for the propagation of drought, and climate change and human
activities are important factors among them. In this paper, we analyzed the effects of temperature, specific
humidity, wind speed, and large-scale climate factors on drought and its propagation. However, there are
numerous and complex factors that affect drought propagation, and different factors interact with each other. It is
necessary to consider the interaction of topography, vegetation coverage, human activities, and climate change,
so as to provide more effective support for drought resistance and control measures.
**Competing interests:**
The authors declare that they have no conflict of interest.
**Author Contributions:**
Min Li (First Author and Corresponding Author): Conceptualization, Methodology, Software, Investigation,
Formal Analysis, Writing-Original Draft;
Zilong Feng: Data Curation, Writing-Original Draft, Writing-Review & Editing;
Mingfeng Zhang: Visualization, Investigation;
Lijie Shi: Superbvision, Validation;
Yuhang Yao: Investigation, Data Curation.
**Acknowledgements.**
We    are    grateful    to    the    the    National    Oceanic    and    Atmospheric    Administration
(http://www.esrl.noaa.gov/psd/data/climateindices) for providing the large climate indices data, and grateful to
the GLDAS (https://disc.gsfc.nasa.gov/datasets/GLDAS_NOAH10_M_2.0/) for providing the average monthly
precipitation, temperature, wind speed, specific humidity, evapotranspiration, soil water content datasets and the
runoff datasets. The data and materials of the research are available.
**Funding Information**
This work was supported by the State Key Laboratory of Hydraulic Engineering Intelligent Construction and
Operation (No. HESS-2206), and the Open Fund of Key Laboratory of Flood & Drought Disaster Defense, the
Ministry of Water Resources (KYFB202307260034).

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
