# Peer review of "Dynamic analysis of drought propagation in the context of"

_Natural Hazards and Earth System Sciences, 2024_

## Referee Comment (RC1)

**General Comments:**

The manuscript examines drought propagation while considering the impact of various meteorological factors and watershed characteristics. This study seeks to expand on existing knowledge of drought propagation by including the effects of a changing environment and watershed features. While many studies have focused on drought propagation using stationary drought indices, research like this, which applies non-stationary drought indices in the context of drought propagation, is increasingly important given the challenges posed by climate change. The manuscript is well-structured and clear, making it of interest to the readers of the NHESS Journal. However, I believe it could be significantly improved by addressing the points mentioned below.

**Review Comments:**

**Major comments:**

L94: As referring to few other articles, the details of area of basin are somewhat different as compared to the ones mentioned in this article. Also, the existing mountainous area and plain area details do not add up to total area of basin (refer article: https://doi.org/10.1016/j.oreoa.2024.100049. )

L98: Could you kindly clarify whether it is possible for a basin to have an annual mean temperature in the range of 1 to 11°C, while the monthly mean temperature ranges from 17 to 25°C?

L119: How you have handled the grid points? Have you thought of considering the influence of the grids just on the boundary of the catchment?

L176: Is there a specific reason why the Gamma distribution has been chosen for evaluating the Standardized Runoff Index (SRI)? While the Gamma distribution is commonly used for the Standardized Precipitation Index (SPI), the appropriate distribution for SRI can vary depending on the characteristics of the catchment. Are there any previous studies that support the use of the Gamma distribution for modelling runoff in this basin? If so, referencing these studies could help justify the decision to use the Gamma distribution. Try considering other distributions also, to improve the accuracy of the results when assessing drought severity and propagation.

L194: You may consider mentioning that these meteorological variables are used as covariates in the evaluation of the non-stationary hydrological drought index. Additionally, could you include a justification for why large-scale climate factors were chosen as covariates in the evaluation of the non-stationary meteorological index, as well as the rationale for using meteorological variables in assessing the non-stationary hydrological index?

L286: Table7: The AIC, SBC, and GD values for the all the different runoff models in the CDS region can be presented first, with the optimal model highlighted in bold, as done in the case of precipitation (Table 5).

**Minor Comments:**

L11: What do you mean by "The analysis of **law of drought propagation**"?

L23: Rephrase "Furthermore, watershed characteristics also be factors influencing spatial differences

in drought propagation." Also, one-two lines about watershed characteristics can also be added in abstract if possible.

L58-L59: Rephrase ("Under the influence…complex and urgent.")

L84-L85: Rephrase ("Furthermore…characteristics"), grammatical mistake

L85-L89: Rephrase ("To assess…respectively"), sentence too long. Improving it will enhance readability

L91: Section 2: more citations should be added in paragraph first, second and third (mainly line 94, 99, 109)

L128: The reference to the figure is incorrect. It should be: 'Figure 2 summarizes the steps of the current study.'

L163: Here, $\alpha$ and $\beta$ represent the shape and scale parameters, respectively, rather than the scale and shape parameters.

L166: It can be mentioned that this approximate conversion is provided by Abramowitz and Irene (1965).

L190: As commented for L163, here, $\alpha$ and $\beta$ represent the shape and scale parameters, respectively, rather than the scale and location parameter of the gamma distributions.

L256: The reference to the table is wrong. It should be: 'Table 4'

L272: Typo error: Fig.4

L285: There can be other combinations of T, H, and W, such as W and H. Could you please include a statement explaining why only these specific model situations were considered?

L338: Could you elaborate on how and why an increase in temperature may be the primary factor contributing to the increased severity of hydrological droughts in the spring under non-stationary conditions, in contrast to stationary conditions?

L349: Rephrase for better clarity ("This…abundant"). Additionally, it would be helpful to mention the seasons varying with months in the study area section, providing readers with a better understanding of the different seasons in the Luanhe River Basin, China. It would also be beneficial to include some details about the general conditions of precipitation, temperature, and humidity across the various seasons.

L358-L360: Why might temperature (T) and humidity (H) be the primary climate-influencing factors in the upstream and downstream areas, in contrast to the midstream area? It may be helpful to include a brief explanation for a general audience to further clarify this.

L406 and L408: The reference to the figure should be consistent throughout the article, using either 'Fig.' or 'Figure' in both instances.

---

## Author Comment (AC1)

General Comments:

The manuscript examines drought propagation while considering the impact of various meteorological factors and watershed characteristics. This study seeks to expand on existing knowledge of drought propagation by including the effects of a changing environment and watershed features. While many studies have focused on drought propagation using stationary drought indices, research like this, which applies non-stationary drought indices in the context of drought propagation, is increasingly important given the challenges posed by climate change. The manuscript is well-structured and clear, making it of interest to the readers of the NHESS Journal. However, I believe it could be significantly improved by addressing the points mentioned below.

Review Comments:

Major comments:

L94: As referring to few other articles, the details of area of basin are somewhat different as compared to the ones mentioned in this article. Also, the existing mountainous area and plain area details do not add up to total area of basin (refer article: https://doi.org/10.1016/j.oreoa.2024.100049.)

**Response:** Thank you very much for your review and valuable comments on our manuscript. According to your suggestion, the area of the Luanhe River Basin is 44,750 km$^2$, of which 43,940 km$^2$ are mountainous areas, as determined by searching authoritative information. The relevant content of the paper has been modified. The modifications are marked in red font. The details are as follows:

The area of the basin is about 44750 km$^2$, with an average width of 90km from east to west and a length of 500km from north to south, including a mountainous area of 43940 km$^2$.

L98: Could you kindly clarify whether it is possible for a basin to have an annual mean temperature in the range of 1 to 11°C, while the monthly mean temperature ranges from 17 to 25°C?

**Response:** Thank you very much for your review and valuable comments on our manuscript. Line 98 of the text should read: The annual mean temperature in the watershed ranges from 1 to 11°C, while the July mean temperature ranges from 17 to 25°C. The relevant text has been amended.

L119: How you have handled the grid points? Have you thought of considering the influence of the grids just on the boundary of the catchment?

**Response:** Thank you very much for your review and valuable comments on our manuscript. In this study, the downloaded grid point data were screened based on the watershed boundaries, and the grid point data whose grid center points were located within the watershed boundaries were selected as study data.

L176: Is there a specific reason why the Gamma distribution has been chosen for evaluating the Standardized Runoff Index (SRI)? While the Gamma distribution is commonly used for the Standardized Precipitation Index (SPI), the appropriate distribution for SRI can vary depending on the characteristics of the catchment. Are there any previous studies that support the use of the Gamma distribution for modelling runoff in this basin? If so, referencing these studies could help justify the decision to use the Gamma distribution. Try considering other distributions also, to improve the accuracy of the results when assessing drought severity and propagation.

**Response:** Thank you very much for your review and valuable comments on our manuscript. In this paper, before constructing the non-stationary drought index, the Weibull, Gamma, Log-normal, and Gumbel four distributions were selected to simulate the runoff. Evaluation of the simulation results based on the AIC, SBC, and GD metrics showed that the Gamma distribution performed optimally in the spring, summer, and autumn seasons, and also performed well in fitting the winter runoff series. Therefore, the Gamma distribution was used to calculate the SRI.

L194: You may consider mentioning that these meteorological variables are used as covariates in the evaluation of the non-stationary hydrological drought index. Additionally, could you include a justification for why large-scale climate factors were chosen as covariates in the evaluation of the nonstationary meteorological index, as well as the rationale for using meteorological variables in assessing the non-stationary hydrological index?

**Response:** Thank you very much for your review and valuable comments on our manuscript. The main purpose of this study is to investigate the influence of factors other than precipitation and runoff on drought propagation. Previous studies have shown that the large-scale climate index is an important factor influencing meteorological drought in the Luan River Basin, and that large-scale climate indices influence runoff indirectly by modulating atmospheric circulation and regional meteorological conditions, while meteorological factors (temperature, specific humidity, wind speed) influence runoff more directly through physical processes directly involved in the water cycle (e.g., precipitation, evaporation, snowmelt), as evidenced by the study of Das et al (2022). Therefore, in this study, large-scale climatic factors were selected as covariates to construct a non-stationary meteorological drought index, and meteorological variables were selected as covariates to construct a non-stationary hydrological drought index.

Das S, Das J and Umamahesh N V. Investigating the propagation of droughts under the influence of large-scale climate indices in India, Journal of Hydrology, 610, 127900. https://doi.org/10.1016/j.jhydrol.2022.127900, 2022.

L286: Table7: The AIC, SBC, and GD values for the all the different runoff models in the CDS region can be presented first, with the optimal model highlighted in bold, as done in the case of precipitation (Table 5).

**Response:** Thank you very much for your review and valuable comments on our manuscript. In this paper, the AIC, SBC and GD values of all runoff models were also calculated for the optimization of different runoff models. Due to excessive data, the AIC, SBC and GD values of all models were not

listed in the table for the sake of making the article more concise. We present the results of all models in the table below:

Table1 AIC, SBC, and GD of the different models of runoff in the CDS region (the Bold indicates the optimal model)

| | Spring | | | Summer | | | Autumn | | | Winter | | |
| --- | --- | --- | --- | --- | --- | --- | --- | --- | --- | --- | --- | --- |
| | AIC | SBC | GD | AIC | SBC | GD | AIC | SBC | GD | AIC | SBC | GD |
| Mod1 | -63.96 | -60.02 | -67.96 | 139.83 | 143.77 | 135.83 | -54.77 | -50.83 | -58.77 | -422.75 | -418.81 | -426.75 |
| Mod2 | -63.66 | -55.78 | -71.66 | 137.69 | 245.57 | 129.69 | -51.45 | -43.57 | -59.45 | -420.46 | -412.57 | -428.46 |
| Mod3 | -69.83 | -61.95 | -77.83 | 142.67 | 150.55 | 134.67 | **-57.83** | **-49.95** | **-65.83** | **-432.39** | **-424.51** | **-440.39** |
| Mod4 | -62.04 | -56.13 | -68.04 | **136.18** | **142.09** | **130.18** | -53.40 | -47.49 | -59.40 | -421.02 | -415.11 | -427.02 |
| Mod5 | -63.53 | -57.62 | -69.53 | 141.62 | 147.53 | 135.62 | 52.81 | -46.90 | 58.81 | -427.48 | -421.57 | -433.48 |
| Mod6 | -65.04 | -59.13 | -71.04 | 140.72 | 146.63 | 134.72 | -52.81 | -46.90 | -58.81 | -421.99 | -416.08 | -427.99 |
| Mod7 | -65.44 | -59.53 | -71.44 | 141.20 | 147.11 | 135.20 | -56.65 | -50.74 | -62.65 | -420.79 | -414.88 | -426.79 |
| Mod8 | -62.02 | -56.11 | -68.02 | 139.86 | 145.77 | 133.86 | -52.77 | -46.86 | -58.77 | -420.86 | -414.96 | -426.86 |
| Mod9 | -61.81 | -51.96 | -71.81 | 138.55 | 148.40 | 128.55 | -49.47 | -39.61 | -59.47 | -418.46 | -408.60 | -428.46 |
| Mod10 | **-70.49** | **-60.64** | **-80.49** | 144.33 | 154.19 | 134.33 | -56.00 | -46.15 | -66.05 | -432.10 | -422.24 | -442.10 |
| Mod11 | -62.91 | -57.00 | -68.91 | 141.60 | 147.51 | 135.60 | -53.34 | -47.43 | -59.34 | -422.49 | -416.58 | -428.49 |
| Mod12 | -66.27 | -58.39 | -74.27 | 142.80 | 150.68 | 134.80 | -55.29 | -47.41 | -63.29 | -420.55 | -412.67 | -428.55 |
| Mod13 | -62.03 | -54.15 | -70.03 | 143.45 | 151.33 | 135.45 | -51.35 | -43.47 | -59.35 | -425.67 | -417.79 | -433.67 |

Minor Comments:

L11: What do you mean by "The analysis of law of drought propagation"?

**Response:** Thank you very much for your review and valuable comments on our manuscript. Here 'analysis of law of drought propagation' specifically refers to revealing the influence of large-scale climate indices and meteorological factors on drought propagation by quantifying the probabilities and thresholds for the propagation of meteorological droughts into hydrological droughts in different

seasons under both stationary and non- stationary conditions. The content here has been revised in response to another reviewer's comments, as follows:

Investigating the processes governing drought propagation under a changing environment is essential for advancing drought early warning and reducing socio-economic risks.

L23: Rephrase "Furthermore, watershed characteristics also be factors influencing spatial differences in drought propagation." Also, one-two lines about watershed characteristics can also be added in abstract if possible.

**Response:** Thank you very much for your review and valuable comments on our manuscript. Relevant contents have been revised based on expert comments and specific dimensions of watershed characteristics (geomorphology and vegetation) have been enumerated.

Furthermore, the spatial variability of drought propagation is further influenced by watershed characteristics, including the slope and leaf area index, which collectively alter runoff generation processes.

L58-L59: Rephrase ("Under the influence…complex and urgent.")

L84-L85: Rephrase ("Furthermore…characteristics"), grammatical mistake

L85-L89: Rephrase ("To assess…respectively"), sentence too long. Improving it will enhance readability

**Response:** Thank you very much for your review and valuable comments on our manuscript. The relevant contents of the paper have been checked and modified according to expert opinions, and the modified contents are marked in red font in the article as follows:

L58-L59: Under the influence of climate change and human activities, precipitation and runoff series show significant non-stationarity and uncertainty, and drought studies become more complex and urgent (Wang et al. 2015; Wang et al. 2020; Jehanzaib et al. 2023).

L84-L85: Furthermore, spatial and temporal differences in drought propagation are strongly related to watershed characteristics.

L85-L89: To evaluate the influence of external driving factors on drought propagation, NSPI and NSRI were constructed based on the GAMLSS framework with climate indices and meteorological factors as covariables. The propagation probability and propagation threshold of meteorological drought to hydrological drought were calculated by the Copula model under stationary and non-stationary conditions in different seasons, respectively.

L91: Section 2: more citations should be added in paragraph first, second and third (mainly line 94, 99, 109)

**Response:** Thank you very much for your review and valuable comments on our manuscript. Citations have been added to lines 94 and 99, respectively, in accordance with the expert's recommendations, while the content of line 109 has been deleted.

L128: The reference to the figure is incorrect. It should be: 'Figure 2 summarizes the steps of the current study.'

**Response:** Thank you very much for your review and valuable comments on our manuscript. The relevant contents of the paper have been checked and modified according to expert opinions, and the modified contents are marked in red font in the article as follows:

Fig.2 summarizes the steps of the current study.

L163: Here, α and β represent the shape and scale parameters, respectively, rather than the scale and shape parameters.

**Response:** Thank you very much for your review and valuable comments on our manuscript. The relevant contents of the paper have been checked and modified according to expert opinions. Modified as follows:

L163: In the formula, $\alpha$ and $\beta$ are shape and scale parameters ($\alpha>0$, $\beta>0$) and they are treated as constants in the GAMLSS framework.

L166: It can be mentioned that this approximate conversion is provided by Abramowitz and Irene (1965).

**Response:** Thank you very much for your review and valuable comments on our manuscript. Added on line 166 on the basis of the experts' suggestion, as follows:

The cumulative probability normalization method is based on the inverse normal function algorithm proposed by Abramowitz and Stegun (1965).

L190: As commented for L163, here, $\alpha$ and $\beta$ represent the shape and scale parameters, respectively, rather than the scale and location parameter of the gamma distributions.

**Response:** Thank you very much for your review and valuable comments on our manuscript. The relevant contents of the paper have been checked and modified according to expert opinions, and the modified contents are marked in red font in the article as follows:

In the formula, $\alpha$ and $\beta$ are shape and scale parameters ($\alpha>0$, $\beta>0$) and they are treated as constants in the GAMLSS framework.

L256: The reference to the table is wrong. It should be: 'Table 4'

L272: Typo error: Fig.4

**Response:** Thank you very much for your review and valuable comments on our manuscript. The relevant contents of the paper have been checked and modified according to expert opinions, and the modified contents are marked in red font in the article as follows:

L256: The AIC, SBC, and GD were used to select the optimal model, taking the CDS region as an example. The results of model preferences for the precipitation series are shown in Table 4.

L272: It can be seen from Fig.4 that the precipitation data values of the four seasons were basically within the 95% quantile interval, the deviation values in the worm chart were evenly distributed in the 95% confidence interval, and there was no obvious excess, which indicates that the residual fitting of the Gamma distribution meets the conditions.

L285: There can be other combinations of T, H, and W, such as W and H. Could you please include a statement explaining why only these specific model situations were considered?

**Response:** Thank you very much for your review and valuable comments on our manuscript. In this paper, 13 models are considered for calculating the optimal model for the runoff sequence, including the combination of T, H, and W, the combination of H and W, the combination of T and W, etc. The models mentioned in the paper are the optimal models for each region. To minimize redundancy, models other than the optimal model are not listed in Table 6.

L338: Could you elaborate on how and why an increase in temperature may be the primary factor contributing to the increased severity of hydrological droughts in the spring under non-stationary conditions, in contrast to stationary conditions?

**Response:** Thank you very much for your review and valuable comments on our manuscript. Based on the model simulation results, the optimal model for most of the regional runoff is Mod3 in Table6, with the covariates in Mod3 being temperature and humidity. According to the results in Table2, the temperature in the watershed shows a significant increasing trend in different seasons, while there is no significant trend in humidity. Therefore, the paper mentions that the increase in temperature is the main factor which leads to the increase in the severity of hydrologic drought in spring. In addition, the optimal model of runoff series from each area of the watershed for the different seasons was added to Section 4.2.2 of the article and is shown in Table 9. Then, changes were also made to Table 6 and Table 7 where relevant. The details are as follows:

Table 6 Different model situations considered for runoff simulation

| Model | Parameter | |
|---|---|---|
| | $\alpha_t$ | $\beta_t$ |
| Mod 1 | ~1 | ~1 |
| Mod 2 | ~1 | ~ T and H |
| Mod 3 | ~ T and H | ~1 |
| Mod 4 | ~1 | ~ T |
| Mod 5 | ~ T | ~1 |
| Mod 6 | ~1 | ~ H |
| Mod 7 | ~ H | ~1 |
| Mod 8 | ~1 | ~W |
| Mod 9 | ~1 | ~ T, H and W |
| Mod 10 | ~ T, H and W | ~1 |
| Mod 11 | ~W | ~1 |
| Mod 12 | ~H and W | ~1 |
| Mod 13 | ~ T and W | ~1 |

Table 7 AIC, SBC, and GD of the best suitable model of the non-stationary model of runoff in the CDS region

| Season | The optimal model | |
|--------|-------------------|---|
| Spring | Mod 10 | AIC: -70.28 |
| | | SBC: -60.43 |
| | | GD: -80.27 |
| Summer | Mod 4 | AIC:136.19 |
| | | SBC:142.10 |
| | | GD:130.19 |
| Autumn | Mod 3 | AIC: -58.67 |
| | | SBC: -50.79 |
| | | GD: -66.67 |
| Winter | Mod 3 | AIC: -4477 |
| | | SBC: -439.89 |
| | | GD: -455.77 |

The results of the optimal modeling of the non-stationary series for the 11 regional runoffs are presented in Table 9. Table 9 shows that there are some differences in the optimal models of the non-stationary runoff series in different regions in different seasons, among which the spatial differences of the optimal models in winter are the most significant. The optimal models in spring are mainly Mod3 and Mod10, in summer the optimal models are mainly Mod2, Mod4 and Mod8, in autumn the optimal models are Mod3 and Mod7, and in winter the optimal models are mainly Mod3 and Mod11.

Table 9 Optimal model of non-stationary runoff series in different seasons in each region of the basin

| Region | Spring | Summer | Autumn | Winter |
|--------|--------|--------|--------|--------|
| ZL | Mod3 | Mod6 | Mod7 | Mod11 |
| DL | Mod3 | Mod2 | Mod7 | Mod11 |
| GY | Mod3 | Mod8 | Mod7 | Mod11 |

| | | | | |
|---|---|---|---|---|
| FN | Mod3 | Mod2 | Mod7 | Mod12 |
| WC | Mod3 | Mod8 | Mod7 | Mod7 |
| LH | Mod3 | Mod4 | Mod3 | Mod3 |
| LP | Mod3 | Mod4 | Mod3 | Mod3 |
| CDS | Mod10 | Mod4 | Mod3 | Mod3 |
| CDX | Mod3 | Mod4 | Mod3 | Mod10 |
| PQ | Mod10 | Mod4 | Mod3 | Mod3 |
| KC | Mod10 | Mod4 | Mod3 | Mod3 |

L349: Rephrase for better clarity ("This…abundant"). Additionally, it would be helpful to mention the seasons varying with months in the study area section, providing readers with a better understanding of the different seasons in the Luanhe River Basin, China. It would also be beneficial to include some details about the general conditions of precipitation, temperature, and humidity across the various seasons.

**Response:** Thank you very much for your review and valuable comments on our manuscript. The percentage of precipitation at different seasons is detailed in the study area section of the article. The details are given below:

Affected by the continental monsoon climate, the basin has four distinct seasons of precipitation, with an average annual precipitation of 400~800mm, of which summer precipitation accounts for 67%-76% of the total annual precipitation; spring and autumn account for about 9% and 15% respectively; and winter precipitation accounts for only about 2% (Li et al. 2023).

L358-L360: Why might temperature (T) and humidity (H) be the primary climate-influencing factors in the upstream and downstream areas, in contrast to the midstream area? It may be helpful to include a brief explanation for a general audience to further clarify this.

**Response:** Thank you very much for your review and valuable comments on our manuscript. Table 9 added to the paper shows the optimal model for the series of unsteady runoff in different seasons in each region of the Luan River Basin. According to Table 9, the optimal model of runoff in ZL, DL, GY, FN, and WC regions is Mod7, and the optimal model of runoff in LH, LP, CDS, CDX, PQ, and KC regions is Mod3. Specific humidity is the main influencing factor in upstream region, and temperature and humidity are the main influencing factors in middle and downstream regions. According to the results in the table all the relevant contents in the text are modified as follows:

As can be seen from Table 9, specific humidity is the main influence on the differences in drought propagation in the upstream (ZL, DL, GY, FN, and WC), while drought in the middle and lower reaches (LH, LP, CDS, CDX, PQ, and KC) is influenced by a combination of temperature and specific humidity.

L406 and L408: The reference to the figure should be consistent throughout the article, using either 'Fig.' or 'Figure' in both instances.

**Response:** Thank you very much for your review and valuable comments on our manuscript. The figure is referenced throughout the article using "Fig." for consistency. Relevant changes to the paper have been highlighted in red.

---

## Author Comment (AC2)

Reviewer2:

This manuscript presents a statistical analysis of drought propagation from meteorological to hydrologic drought. This is certainly an important topic and fits within the scope of the journal. I have some major comments that need to be addressed before publication that relate to the lack of validation against drought observations, the suitability of what is considered a measure of hydrologic drought as well as the clarity in description of the methods.

Major comments:

Drought Definitions including Hydrologic Drought and Model Validation:

I have some major questions about the applicability of the method. For example, I tend to use SM percentiles or Evaporative Stress Index as drought indicators. In my world it is typical to estimate a distribution of values for a given time period (e.g. week of year and then to look at the percentile anomaly). Here is seems as if there is a single underlying distribution for the entire year (or season), which is then evaluated. I am just wondering how well the model actually represents something that is observable as drought. This could easily be validated by looking at observed conditions. The authors determine that there are differences between drought propagation between both models, but are the underlying models actually good models for real world drought conditions rather than good models for fitting a precipitation distribution. I am also a bit confused by the definition of hydrologic drought, which seems to be based on surface runoff from the Noah model. Typically hydrologic drought is assessed from gauged river runoff, while surface runoff is a temporary phenomenon occurring after strong precipitation events. I am not sure whether it makes sense to use the Noah gridded runoff as an indicator of hydrologic drought in the first place. How does this compare to for example river gauges? Again this could be validated against observations.

**Response:** Thank you very much for your review and valuable comments on our manuscript. SPI and SRI are commonly used to characterize meteorological and hydrological drought, and can characterize drought at multiple event scales, including 1-month, 3-month, and 12-month time scales, with the different time scales based on different cumulative rainfall and runoff series. The 1-month

time scale is often used to represent monthly droughts, the 3-month time scale is often used to characterize seasonal droughts, and the 12-month time scale is often used to characterize annual droughts. The main objective of this study is to investigate the influence of large-scale climate indices and regional meteorological factors on the propagation characteristics of meteorological drought to hydrological drought in different seasons. Therefore, this paper is based on the drought indices of 3-month time scale.

As stated by the experts, the runoff data used in this paper are runoff based on the Noah model. The aim of this paper is to investigate the spatial differences in the characteristics of drought propagation in the Luanhe River Basin by using large-scale climate indices and regional meteorological factors. Due to the lack of measured data and the small number of stations, the Noah model's data have high resolution and continuous time series. Therefore, the Noah model data were used in this study and the SRI based on Noah model data also performed well in capturing drought events in typical years in the basin. In addition, the study focuses on comparing the differences in drought propagation between stationary and non-stationary scenarios.

Methods Description:

In general I find that methods need to be described better for the reader to understand how exactly the models are formulated. It is for example not clear to me what specifically is meant by non-stationary model. Non stationary with respect to what (climate change trends, seasonal cycles, teleconnections) etc.

**Response:** Thank you very much for your review and valuable comments on our manuscript. The content of the Methods section of the article was optimized and modified according to the experts' comments.

Generalized Additive Models for Location, Scale, and Shape (GAMLSS) proposed by Rigby and Stasinopoulos (2005)can flexibly analyze non-stationary time series, more details of GAMLSS are available in Rigby et al. (2005). In recent years it has often been applied to capture non-stationary

in series such as precipitation and runoff. The non-stationary model presented in this paper is based on the study by Das et al. (2022). To better study the seasonal characteristics of drought and capture the changes in meteorological elements caused by seasonal climate change, this paper chooses the drought index on a 3-month time scale to analyze the propagation characteristics of drought, and the GAMLSS model is used to construct a non-stationary model for the analysis of precipitation and runoff changes. By incorporating large-scale climate factors as covariates, a non-stationary meteorological drought index is constructed and used to capture the non-stationary characteristics of precipitation series in the basin. In this paper, based on the calculation principle of the standardized drought index, the non-stationary Gamma distribution of precipitation and runoff is constructed based on the GAMLSS model. The correlated climate variables are selected from these large-scale climate factors (e.g., AMO, SOI, PDO, AO, NAO, and NP). To capture the non-stationary characteristics of the basin runoff sequence, the non-stationary hydrological drought index (NSRI) was constructed. The meteorological variables (wind speed, temperature, and specific humidity) were considered as covariates for the non-stationary model. The semiparametric additive model formula used in this study is as follows:

$$g_1(\alpha_t) = \sum_{j=1}^{j_k} h_{jk}(c_{jk}) \tag{3}$$

$$g_2(\beta_t) = \sum_{j=1}^{j_k} h_{jk}(c_{jk}) \tag{4}$$

Where $g_1(\alpha_t)$ is the link function, which is determined by the domain of the statistical parameter, namely, if the domain of the distributed parameter $\alpha_t$ is $\alpha_t \in R$, the link function is $g_1(\alpha_t) = \alpha_t$, if $\alpha_t > 0$, then $g_1(\alpha_t) = \ln \alpha_t$. The $h_{jk}$ represents the dependence function of the distribution parameters on the covariates $c_{jk}$. The parameter coefficients and model residuals are estimated by RS algorithm, and whether the model residuals approximately satisfy the normal distribution is analyzed, and the optimal fitting distribution is selected by AIC (Akaike information criterion), SBC (Schwarz Bayesian Criterion), and GD (Global Deviance).

Presentation and Interpretation

In general figures and tables should have captions that clearly help with interpretation of the

information presented there. This should be accompanied by a description and interpretation of the figure content.

**Response:** Thank you very much for your review and valuable comments on our manuscript. The titles of the figures and tables in the text have been checked. Figs.4 and 5 have been annotated to clearly explain the meaning of the various symbols in the figures.

Specific Comments

L22: "propagation thresholds in winter significantly increasing by 0.1-0.2" > it is unclear hear what these thresholds represent. I assume that is explained in the text, but with the abstract information alone it is to me not comprehensible.

**Response:** Thank you very much for your review and valuable comments on our manuscript. The threshold in "Propagation threshold in winter significantly increases by 0.1-0.2" is defined as the severity of meteorological drought (value of SPI) that is most likely to trigger hydrological drought (SRI<-0.5). The definition of the drought propagation threshold is described in Section 3.2, "The Copula model", of the main text.

L31: "evolution from one drought to another is called drought propagation " > I would reformulate this because one can also think of spatial drought propagation where drought propagates from one region to another. I recommend to find a terminology that is less ambiguous.

**Response:** Thank you very much for your review and valuable comments on our manuscript. "Propagation" often implies spatial dispersion in drought studies. This paper, however, focuses on the process of meteorological conditions being accumulated over time and transformed accordingly into hydrological drought, which is drought propagation between drought types. Therefore, "evolution from one drought to another is called drought propagation" in Line31 is modified as follows:

This process of meteorological drought triggering hydrological or agricultural drought is called inter-type drought propagation (Zhang et al. 2021; Wossenyeleh et al. 2021; Apurv and Cai 2020; Jehanzaib et al. 2020).

L98: "he annual mean temperature ranges from 1 to 11°C, and the monthly mean temperature ranges from 17 to 25°C. " > this requires explanation as the two ranges don't seem to go together unless one represents elevation changes and the other monthly mean average over the basin. In any case information about what is averaged over should be given.

**Response:** Thank you very much for your review and valuable comments on our manuscript. Line 98 of the text should read: The annual mean temperature in the watershed ranges from 1 to 11°C, while the July mean temperature ranges from 17 to 25°C. The relevant text has been amended.

L105:" With global climate change, drought disasters in the Luanhe River Basin are becoming increasingly frequent," > the increasing frequency is not really apparent in the list of major drought events given in L108. Please explain. Also, one has to be careful here given that there have also been major changes in land-use and other conditions which would probably dominate economic losses rather than climate change. This is not to say that climate change is not contributing, but I don't think that the information presented supports climate change as a reason or even increasing frequency.

**Response:** Thank you very much for your review and valuable comments on our manuscript. I thank the reviewers for their careful observations. The long-calendar drought events (L108) listed in the paper are mainly used to characterize historical extreme drought cases, while the expression 'increased drought frequency' focuses more on the increase in short-calendar, seasonal droughts. The decrease in long-calendar-time droughts coexists with the frequency of short-calendar-time droughts due to water conservancy projects that enhanced the drought-resistant capacity of the watersheds, which is in line with the characterization of drought differentiation under the combined effects of human activities and climate change. For example, the frequency of drought events extracted based

on the SRI index was 0.66 and 0.82 for the CDS region during 1961-1979 and 1980-2014, respectively.

We fully agree with the reviewers that land use changes (e.g., soil and water conservation) and hydraulic projects (e.g., Panjiakou Reservoir) in the Luanhe River Basin significantly influence the basin's drought evolution characteristics by regulating runoff. Climate change is also an important factor affecting drought in the Luanhe River Basin. Therefore, a study on the impact of climate change on drought in the Luanhe River Basin has been added as a testimony to the summary part of the text. The details are as follows:

As the main source of water supply for the Beijing-Tianjin-Tangshan area, the Luanhe River Basin is responsible for multiple tasks such as urban water supply, and industrial and agricultural water supply. Frequent droughts in recent years have not only affected the supply of regional water resources but also had a serious impact on the ecological environment. Therefore, an in-depth understanding of the evolution pattern and impact mechanism of drought is of great significance to the rational allocation of water resources and sustainable development of the basin. According to some recent studies, there are nonstationary characteristics in the precipitation series and the runoff series of the Luanhe River Basin (Li et al. 2019a; Li et al. 2020). And the occurrence of drought in Luanhe River Basin may be related to some large-scale climatic indices (Wang et al. 2018; Li et al. 2015; Wang et al. 2016). Wang et al. (2016) pointed out that the Atlantic Multi-Year Oscillation (AMO) has a significant effect on drought in the Luan River Basin. In addition, meteorological factors are also important in influencing droughts in the basin. Chen et al. (2022) pointed out that the increase in temperature will lead to more frequent hydrological droughts in the Luan River basin in the future. Previous studies on the Luanhe River Basin have focused on examining the effects of large-scale climatic factors on a single type of drought, with few assessments of the effects of large-scale climatic indices and regional meteorological elements on drought propagation (Li et al. 2015; Wang et al. 2015; Li et al. 2024).

L118 "https://disc.gsfc.nasa.gov/datasets/GLDAS_NOAH10_M_2.0/" > This link is no longer working. Given the absolute mess that the USA are currently experiencing. Rather than providing links dataset IDs should be used that identify datasets conclusively.

**Response:** Thank you very much for your review and valuable comments on our manuscript. The majority of publications using GES DISC data do NOT include the data version ID. Therefore, the link to the dataset is still provided in this paper. We have updated the data links so that you can jump directly to the data URLs through the links. The details are as follows:

The average monthly precipitation, temperature, wind speed, specific humidity, evapotranspiration, and runoff datasets are available at a grid resolution of 0.25° Lat × 0.25° Lon and are obtained from Global Land Data Assimilation System (GES DISC Dataset: GLDAS Noah Land Surface Model L4 monthly 0.25 x 0.25 degree V2.0 (GLDAS_NOAH025_M 2.0) ).

L121: "https://daac.ornl.gov/" > not a dataset link but a link to a data center (see comment above).

**Response:** Thank you very much for your review and valuable comments on our manuscript. We have modified the dataset links in the text based on expert opinions as follows:

The grid-wise analysis is carried out at a resolution of 0.25° Lat × 0.25° Lon over the Luanhe River which includes 58 grid points (Fig. 1(b)). Leaf area index of 0.25° spatial resolution was derived from the Advanced Very High Resolution Radiometer (AVHRR) Global Inventory Modeling and Mapping Studies (GIMMS) LAI3g version 2 (https://daac.ornl.gov/cgi-bin/dsviewer.pl?ds_id=1653 ) (1981–2015).

Figure 1: How are these sub-regions defined. Are these sub watersheds or corresponding to administrative/ municipal districts? For a propagation analysis, it would make sense to divide by subwatersheds rather than administrative divisions.

**Response:** Thank you very much for your review and valuable comments on our manuscript. The 11 sub-regions in Fig.1 are organized according to the administrative divisions contained in the watershed. We fully agree with the reviewers that the analysis of drought propagation based on sub-watershed delineation is more meaningful than administrative divisions. Although administrative boundaries were used in this study, the study also considered the effects of watershed characteristics such as slope and land use type of subregions on drought propagation. Future research will further integrate multi-scale analysis of natural sub-basins and administrative units to fully reveal the driving mechanisms of drought propagation.

Section3/ Figure 2: If the figure is shown at the beginning of the section it should be explained in the text right then and there. It is also not clear to me whether there is really a flow between Step 1 and Step 2 since both start from data rather than Step 2 using Step 1 outputs. This figure should be revised accordingly.

**Response:** Thank you very much for your review and valuable comments on our manuscript. The explanation in Fig. 2 has been added at the beginning of section III on the basis of the comments of the experts. The details are as follows:

The current study aims to assess the impact of external drivers on drought propagation based on the GAMLSS model, in particular, the probability and threshold of drought propagation in different seasons. Fig.2 summarizes the steps of the current study. First, the standardized drought indices SPI and SRI were calculated based on monthly-scale rainfall and runoff. Then, the non-stationary drought indices NSPI and NSRI were calculated using the large-scale climate indices and regional meteorological factors as covariates of the non-stationary Gamma distribution parameters of the rainfall and runoff series, respectively. Finally, based on the Copula function, the drought was computed under stationary and non-stationary conditions, respectively. Transmission characteristics based on Copula function to analyze and quantify the influence of external drivers on the propagation of meteorological drought to hydrological drought.

With regard to "It is also not clear to me whether there is really a flow between Step 1 and Step 2 since both start from data rather than Step 2 using Step 1 outputs. This figure should be revised accordingly." Steps 1 and 2 in the text indicate the sequence in which the study was carried out, and the non-stationary aridity index constructed in Step 2 is based on the calculation of the standardized aridity index in Step 1. Step 3, on the other hand, is based on the aridity index calculated in Steps 1 and 2. In addition, the steps are laid out in such a way as to consider the aesthetics of the figure.

Section 3.1: Given that Pearson correlation is a standard method, I don't think it needs to be explained in detail. It would increase conciseness and legibility of the manuscript to remove this section and to briefly mention it when the correlation is applied

**Response:** Thank you very much for your review and valuable comments on our manuscript. The Pearson correlation analysis in section 3.1 has been deleted on the basis of expert advice. It was also used in section 4.1 with a brief reference to the method to increase the brevity and readability of the manuscript. The modifications are described in section 4.1 and the modifications have been highlighted in red.

Section 3.2. If GAMLSS only applies to the non-stationary part it should be introduced as a subsection there. If not it is missing in Step 1 of figure 2. In all cases a better explanation is needed

**Response:** Thank you very much for your review and valuable comments on our manuscript. The GAMLSS is applicable in both stationary and non- stationary models, and Mod1 of the rainfall and runoff models in the text refers to the stationary model (both $\alpha$ and $\beta$ are constants), and the performance of the non-stationary models is judged by comparing with the stationary model. The standardized drought index is calculated directly in Step 1 and does not use the GAMLSS model.

3.2.1: This is the standard approach and one could think of whether this is needed, whereas the non-stationary part is new/ less commonly applied. So the focus on the methods should lie on that.

3.2.2 Non-stationary model.

My expertise is in drought and not necessarily statistics. I feel that this section is too short and I have trouble understanding what specifically is different between the two models. How are "large-scale climate factors as covariates" included into the model. I feel that this requires quite a bit more explanation, since that is the novel/ core aspect of the manuscript.

**Response:** Thank you very much for your review and valuable comments on our manuscript. The structure of the article has been modified accordingly, based on the comments of the experts, as described in section 3.1.2, "Non-stationary model", of the article.

Generalized Additive Models for Location, Scale, and Shape (GAMLSS) proposed by Rigby and Stasinopoulos (2005)can flexibly analyze non-stationary time series, more details of GAMLSS are available in Rigby et al. (2005). In recent years it has often been applied to capture non-stationary in series such as precipitation and runoff. The non-stationary model presented in this paper is based on the study by Das et al. (2022). To better study the seasonal characteristics of drought and capture the changes in meteorological elements caused by seasonal climate change, this paper chooses the drought index on a 3-month time scale to analyze the propagation characteristics of drought, and the GAMLSS model is used to construct a non-stationary model for the analysis of precipitation and runoff changes. By incorporating large-scale climate factors as covariates, a non-stationary meteorological drought index is constructed and used to capture the non-stationary characteristics of precipitation series in the basin. In this paper, based on the calculation principle of the standardized drought index, the non-stationary Gamma distribution of precipitation and runoff is constructed based on the GAMLSS model. The correlated climate variables are selected from these large-scale climate factors (e.g., AMO, SOI, PDO, AO, NAO, and NP). To capture the non-stationary characteristics of the basin runoff sequence, the non-stationary hydrological drought index (NSRI) was constructed. The meteorological variables (wind speed, temperature, and specific humidity) were considered as covariates for the non-stationary model. The semi-parametric additive model formula used in this

study is as follows:

$$g_1(\alpha_t) = \sum_{j=1}^{j_k} h_{jk}(c_{jk}) \qquad (7)$$

$$g_2(\beta_t) = \sum_{j=1}^{j_k} h_{jk}(c_{jk}) \qquad (8)$$

Where $g_1(\alpha_t)$ is the link function, which is determined by the domain of the statistical parameter, namely, if the domain of the distributed parameter $\alpha_t$ is $\alpha_t \in R$, the link function is $g_1(\alpha_t) = \alpha_t$, if $\alpha_t > 0$, then $g_1(\alpha_t) = \ln \alpha_t$. The $h_{jk}$ represents the dependence function of the distribution parameters on the covariates $c_{jk}$. The parameter coefficients and model residuals are estimated by RS algorithm, and whether the model residuals approximately satisfy the normal distribution is analyzed, and the optimal fitting distribution is selected by AIC (Akaike information criterion), SBC (Schwarz Bayesian Criterion), and GD (Global Deviance).

Right now in my limited and very likely incorrect understanding it seems to me that the non-stationary model represents different gamma distributions for 3 month seasons?

**Response:** Thank you very much for your review and valuable comments on our manuscript. A three-month time-scale drought index is commonly used to assess seasonal drought, which is calculated using three-month cumulative precipitation and runoff series as the base data. While traditional stationary models tend to assume that the distribution parameters of the data are fixed, non-stationary models allow the parameters of the distribution (shape parameter and scale parameter in this study) to change dynamically with external covariates (e.g., climate indices, meteorological factors), thus capturing the dynamic evolution pattern of the data distribution in a more flexible way. The optimal parameters of the non-stationary Gamma distribution are identified based on the AIC criterion, etc., and then the drought index is calculated based on the optimal parameters of the distribution.

L232: 'CI', AMO, ... > define these here clearly (and avoid exessive abbreviations)

**Response:** Thank you very much for your review and valuable comments on our manuscript. CI denotes the large-scale climate index and AMO denotes the Atlantic Interdecadal Oscillation, abbreviations that were defined when they first appeared in the manuscript.

Figure 3: Correlation over which time period? Also please provide a clear in text explanation on how to read this figure. What are the main take-aways, I am currently struggling to understand the significance of this figure. So there seems to be a positive correlation between precipitation and a teleconnection) in May, which then flips to negative later in the year. Is there a specific reason to show AMO or is this just to give an example. If just an example than other plots should be shown in supplement.

**Response:** Thank you very much for your review and valuable comments on our manuscript. Figure 3: Correlation of AMO with rainfall series during 1961-2014. In this study, the correlation between the cumulative precipitation series with different time scales and the large-scale climate index was calculated for all regions of the basin, and the results showed that the most relevant large-scale climate index for the cumulative precipitation series with a three-month time scale was the AMO in all 11 regions, and then the correlation between the AMO with different sliding averages for the lead times of 0, 1, 2, and 3 months and the three-month time scale was calculated. Cumulative precipitation series, and the results show that AMO-1 and AMO-24 for the same period have a high correlation with the precipitation series.

Table 2: Trends over which periods?

Table 2 and 4: It would be good to provide a clear comparison of these model parameters to the non-stationary model parameters.

L268: This explanation should be in the figure caption of figure 4.

**Response:** Thank you very much for your review and valuable comments on our manuscript.

Table 2 shows trends in meteorological factors over the period 1961-2014, and the title of Table 2 has been modified.

About "Table 2 and 4: It would be good to provide a clear comparison of these model parameters to the non-stationary model parameters." The stationary models have been considered in the paper when combining the covariates. Mod1 in Tables 3 and 6 is the stationary model, and the results of model preference based on AIC, SBC and GD show that the non-stationary model has a better performance.

About "L268: This explanation should be in the figure caption of figure 4." Relevant explanations have been added to the captions of Figures 4 and 5. The specific modifications are as follows:

Figure 4 Fitting results of four seasons of precipitation series in the CDS region (These red dots represent precipitation observations, light grey areas represent areas between the 5% and 95% centile curves, dark grey areas represent areas between the 25% and 75% centile curves, and black lines represent the median (50%); the black dashed line in the worm plot of the fitted residuals indicates the 95% confidence interval).

Figure 5 Fitting results of four seasons of runoff series in the CDS region (These red dots represent runoff observations, light grey areas represent areas between the 5% and 95% centile curves, dark grey areas represent areas between the 25% and 75% centile curves, and black lines represent the median (50%); the black dashed line in the worm plot of the fitted residuals indicates the 95% confidence interval).

L273: "quantile" > percentile?

**Response:** Thank you very much for your review and valuable comments on our manuscript. The text is specifically "95% quantile interval", because the previous data was "95%", so "quantile" is used.

Figure 4: I don't quite understand what the right column of the plot indicates. Please provide a clear explanation of what is shown and the significance. Does the right side indicate that the fitted gamma function for precipitation is able to reproduce the statistical distribution of precipitation? Also do you

have a comment on the behavior of the fitted gamma functions at the extremes, which are obviously much harder to model with confidence (hence also the larger space of the CI lines).

**Response:** Thank you very much for your review and valuable comments on our manuscript. The black dashed line in the worm plot of the fitted residuals in the right column of Fig. 4 indicates the 95% confidence interval. The text has a relevant explanation: "The deviation values in the worm chart were evenly distributed in the 95% confidence interval, and there was no obvious excess, which indicates that the residual fitting of the Gamma distribution meets the conditions."

Figure 5: Runoff should have units of volume / time not mm. This figure also leads me to a major question about the definition of hydrological drought used here.

**Response:** Thank you very much for your review and valuable comments on our manuscript. Because the runoff data used in this study are from the Global Land Data Assimilation System, the corresponding dataset definition of runoff is runoff depth per unit area. While SRI analyzes the statistical distribution characteristics of runoff, the units (e.g., mm, $m^3/s$) do not affect the standardization results, and the standardization process removes the magnitude of the original data, so that the final results reflect only the degree of deviation of runoff volume relative to the historical mean.

I am stopping here with specific comments, because I have some major general comments that I think need to be addressed.

Technical

L11: "the law of drought propagation" > this seems like a language issue: "processes governing drought propagation"?

L12: meteorological factorS (?)

There is some editing needed on language and also punctuation and spacing. this should either be done during copy-editing or by the authors before the next submission. Examples include L52 [")h" --> ") h"] L61 (, -> .),

L272: Fig 44 > Fig 4

L269: PERcentile?

**Response:** Thank you very much for your review and valuable comments on our manuscript. The relevant content has been revised in the light of the expert comments, and the revisions are highlighted in red in the manuscript.

L11: Investigating the processes governing drought propagation under a changing environment is essential for advancing drought early warning and reducing socio-economic risks.

L12: "meteorological factor" was replaced by "meteorological factors"

L52: For instance, Jehanzaib et al. (2020) and Peña-Gallardo et al. (2019) have found that climate type, climate change, catchment characteristics, and other factors can affect the propagation of drought.

L61: Therefore, researchers incorporate non-stationarity into drought studies through more appropriate analytical tools. The GAMLSS model is one of the commonly used methods.

L272: "Fig 44" was replaced by "Fig. 4"

L269: The language used in this paper to describe this figure has been adapted from other literature, and "centile" is the correct word (Wang et al. 2023).

Wang Y, Peng T, He Y, Singh V P, Lin Q, Dong X, Fan T, Liu J, Guo J and Wang G. Attribution analysis of non-stationary hydrological drought using the GAMLSS framework and an improved SWAT model, Journal of Hydrology, 627, 130420. https://doi.org/10.1016/j.jhydrol.2023.130420, 2023.

---

## Referee Report (RR1)

**General Comments:**

The manuscript explores drought propagation, taking into account various meteorological factors and watershed characteristics. This study aims to expand existing knowledge by incorporating the effects of a changing environment and watershed features. While many studies have focused on drought propagation using stationary drought indices, research like this, applying non-stationary drought indices, is increasingly important in the context of climate change. The manuscript is well-structured and clear, making it of interest to readers of the NHESS Journal. However, there are a few comments that should be addressed to further improve the manuscript:

**Review Comments:**

**Major comments:**

L 128: Fig. 1(c): Please include a paragraph in the Study Area and Data section about the subdivision of the river basin. It's crucial to detail these administrative divisions, specifying which are upstream, midstream, and downstream. You've mentioned considering watershed characteristics such as slope and land use type. It's important to clarify whether these characteristics were applied to the administrative regions or based on watershed boundary subdivisions. Also, please provide information on the sources of data for slope, evapotranspiration, soil moisture content, etc. Adding these details will enhance the clarity and depth of the manuscript.

L 149: Since you've mentioned that summer precipitation accounts for about 70% of the total, zero precipitation events would be common during the non-monsoon period. How did you deal with this? A two-parameter gamma distribution isn't defined for zero values. Have you considered using a mixed cumulative distribution function?

L208-211: In equation (10) for the copula model, you've expanded it in the survival form. Could you clarify the logic behind this? since we are primarily interested with W<=v and Z<=u form. So mentioning it in survival form seems inconsistent with the explanation and equation. Please revisit the equation, as the left-hand side has the CDF form, while the right-hand side uses the survival function form.

L 456: Discussion: The discussion section seems underdeveloped. It would be beneficial to expand it to offer readers valuable insights. Although you've touched on the influence of watershed characteristics on propagation thresholds, the discussion remains somewhat generic. Since this is a key focus of your paper, consider providing a more detailed analysis to fully engage your audience.

Overall, I suggest providing supplementary material that includes additional plots, tables, and figures not included in the main article to maintain conciseness. This will allow interested readers to gain comprehensive information about your study.

**Minor Comments:**

**L 20- 23**: You mentioned "...upstream and midstream regions, with...". You could specify it further by stating "...upstream and midstream regions of the Luanhe River Basin, China...".

L 44: Could you specify if you mean "Markov models"?

L 79: Change "...Basin. And..." to "...Basin and..."

L 115: Please rephrase the sentence: "Under the... more complex," as "evolution law" seems unnecessary in this context.

L 137-140: Rephrase "Finally, based... hydrological drought." Were the drought conditions computed, or were the drought propagation probabilities calculated based on the copula function?

L 145- 146: Rephrase for better clarity "Taking precipitation as the object.. relatively simile calculation"

L171: non stationary? -> non- stationarity

L 183-184: You can clearly mention here that the meteorological variables (wind speed, temp and specific humidity) were considered as covariates for the non- stationary model of hydrological drought index.

L 187: You have explained the link function $g_1(\propto_t)$. You could also add $g_2(\beta_t)$ to this.

L 192: Akaike information criterion -> Akaike **I**nformation **C**riterion

L 197: Since you have mentioned in equation (9) that u and v represent the two variables respectively. This looks strange to me as u and v are the marginal distributions of the two variables rather than the variables itself?

L 203: "…tails. And.."  -> "…tails and.."

L 224: "…conditions." -> "...conditions, respectively."

L 235: As you mentioned, AMO-1 and AMO-24 were selected as covariates for the precipitation series based on the correlation test results. You could add a line explaining the rationale for selecting AMO-1 and AMO-24. Additionally, including the correlation test results for other climate indices across all lead times in the supplementary material would be beneficial.

L287: Table 7: Winter AIC value looks a bit strange.

L 291: "…non-stationary.." -> "…non-stationary model.."

L 324: Could you provide a comparison of SPI and NSPI across different seasons in the CDS region from 1961 to 2014?

L 478: Lead area index -> Leaf area index